# LARGE SCALE GAN TRAINING FOR HIGH FIDELITY NATURAL IMAGE SYNTHESIS

**Andrew Brock**[*][†]
Heriot-Watt University
ajb5@hw.ac.uk

**Jeff Donahue**[†]
DeepMind
jeffdonahue@google.com

**Karen Simonyan**[†]
DeepMind
simonyan@google.com

## ABSTRACT

Despite recent progress in generative image modeling, successfully generating high-resolution, diverse samples from complex datasets such as ImageNet remains an elusive goal. To this end, we train Generative Adversarial Networks at the largest scale yet attempted, and study the instabilities specific to such scale. We find that applying orthogonal regularization to the generator renders it amenable to a simple "truncation trick," allowing fine control over the trade-off between sample fidelity and variety by reducing the variance of the Generator's input. Our modifications lead to models which set the new state of the art in class-conditional image synthesis. When trained on ImageNet at 128×128 resolution, our models (BigGANs) achieve an Inception Score (IS) of 166.5 and Fréchet Inception Distance (FID) of 7.4, improving over the previous best IS of 52.52 and FID of 18.65.

## 1 INTRODUCTION

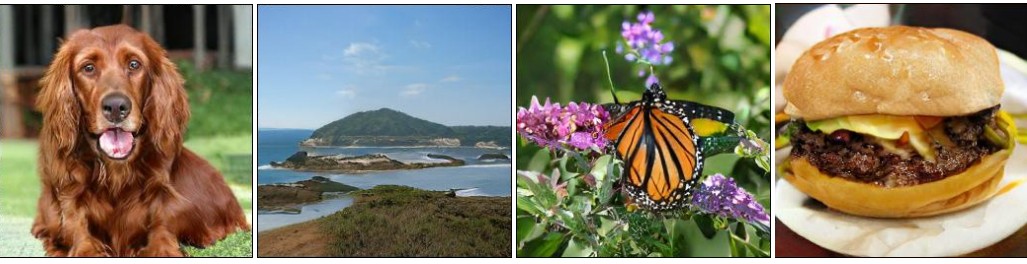

Figure 1: Class-conditional samples generated by our model.

The state of generative image modeling has advanced dramatically in recent years, with Generative Adversarial Networks (GANs, Goodfellow et al. (2014)) at the forefront of efforts to generate high-fidelity, diverse images with models learned directly from data. GAN training is dynamic, and sensitive to nearly every aspect of its setup (from optimization parameters to model architecture), but a torrent of research has yielded empirical and theoretical insights enabling stable training in a variety of settings. Despite this progress, the current state of the art in conditional ImageNet modeling (Zhang et al., 2018) achieves an Inception Score (Salimans et al., 2016) of 52.5, compared to 233 for real data.

In this work, we set out to close the gap in fidelity and variety between images generated by GANs and real-world images from the ImageNet dataset. We make the following three contributions towards this goal:

- We demonstrate that GANs benefit dramatically from scaling, and train models with two to four times as many parameters and eight times the batch size compared to prior art. We introduce two simple, general architectural changes that improve scalability, and modify a regularization scheme to improve conditioning, demonstrably boosting performance.

---

[*]Work done at DeepMind
[†]Equal contribution

- As a side effect of our modifications, our models become amenable to the "truncation trick," a simple sampling technique that allows explicit, fine-grained control of the trade-off between sample variety and fidelity.

- We discover instabilities specific to large scale GANs, and characterize them empirically. Leveraging insights from this analysis, we demonstrate that a combination of novel and existing techniques can reduce these instabilities, but complete training stability can only be achieved at a dramatic cost to performance.

Our modifications substantially improve class-conditional GANs. When trained on ImageNet at 128×128 resolution, our models (BigGANs) improve the state-of-the-art Inception Score (IS) and Fréchet Inception Distance (FID) from 52.52 and 18.65 to 166.5 and 7.4 respectively. We also successfully train BigGANs on ImageNet at 256×256 and 512×512 resolution, and achieve IS and FID of 232.5 and 8.1 at 256×256 and IS and FID of 241.5 and 11.5 at 512×512. Finally, we train our models on an even larger dataset – JFT-300M – and demonstrate that our design choices transfer well from ImageNet. Code and weights for our pretrained generators are publicly available [1].

## 2 BACKGROUND

A Generative Adversarial Network (GAN) involves Generator (**G**) and Discriminator (**D**) networks whose purpose, respectively, is to map random noise to samples and discriminate real and generated samples. Formally, the GAN objective, in its original form (Goodfellow et al., 2014) involves finding a Nash equilibrium to the following two player min-max problem:

$$\min_G \max_D \mathbb{E}_{x \sim q_{\mathrm{data}}(\boldsymbol{x})}[\log D(\boldsymbol{x})] + \mathbb{E}_{\boldsymbol{z} \sim p(\boldsymbol{z})}[\log(1 - D(G(\boldsymbol{z})))], \tag{1}$$

where $\boldsymbol{z} \in \mathbb{R}^{d_z}$ is a latent variable drawn from distribution $p(\boldsymbol{z})$ such as $\mathcal{N}(0, I)$ or $\mathcal{U}[-1, 1]$. When applied to images, **G** and **D** are usually convolutional neural networks (Radford et al., 2016). Without auxiliary stabilization techniques, this training procedure is notoriously brittle, requiring finely-tuned hyperparameters and architectural choices to work at all.

Much recent research has accordingly focused on modifications to the vanilla GAN procedure to impart stability, drawing on a growing body of empirical and theoretical insights (Nowozin et al., 2016; Sønderby et al., 2017; Fedus et al., 2018). One line of work is focused on changing the objective function (Arjovsky et al., 2017; Mao et al., 2016; Lim & Ye, 2017; Bellemare et al., 2017; Salimans et al., 2018) to encourage convergence. Another line is focused on constraining **D** through gradient penalties (Gulrajani et al., 2017; Kodali et al., 2017; Mescheder et al., 2018) or normalization (Miyato et al., 2018), both to counteract the use of unbounded loss functions and ensure **D** provides gradients everywhere to **G**.

Of particular relevance to our work is Spectral Normalization (Miyato et al., 2018), which enforces Lipschitz continuity on **D** by normalizing its parameters with running estimates of their first singular values, inducing backwards dynamics that adaptively regularize the top singular direction. Relatedly Odena et al. (2018) analyze the condition number of the Jacobian of **G** and find that performance is dependent on **G**'s conditioning. Zhang et al. (2018) find that employing Spectral Normalization in **G** improves stability, allowing for fewer **D** steps per iteration. We extend on these analyses to gain further insight into the pathology of GAN training.

Other works focus on the choice of architecture, such as SA-GAN (Zhang et al., 2018) which adds the self-attention block from (Wang et al., 2018) to improve the ability of both **G** and **D** to model global structure. ProGAN (Karras et al., 2018) trains high-resolution GANs in the single-class setting by training a single model across a sequence of increasing resolutions.

In conditional GANs (Mirza & Osindero, 2014) class information can be fed into the model in various ways. In (Odena et al., 2017) it is provided to **G** by concatenating a 1-hot class vector to the noise vector, and the objective is modified to encourage conditional samples to maximize the corresponding class probability predicted by an auxiliary classifier. de Vries et al. (2017) and

---

[1] https://tfhub.dev/s?q=biggan

| Batch | Ch. | Param (M) | Shared | Skip-$z$ | Ortho. | Itr $\times 10^3$ | FID | IS |
|-------|-----|-----------|--------|----------|--------|-------------------|-----|-----|
| 256 | 64 | 81.5 | SA-GAN Baseline | | | 1000 | 18.65 | 52.52 |
| 512 | 64 | 81.5 | ✗ | ✗ | ✗ | 1000 | 15.30 | 58.77($\pm$1.18) |
| 1024 | 64 | 81.5 | ✗ | ✗ | ✗ | 1000 | 14.88 | 63.03($\pm$1.42) |
| 2048 | 64 | 81.5 | ✗ | ✗ | ✗ | 732 | 12.39 | 76.85($\pm$3.83) |
| 2048 | 96 | 173.5 | ✗ | ✗ | ✗ | 295($\pm$18) | 9.54($\pm$0.62) | 92.98($\pm$4.27) |
| 2048 | 96 | 160.6 | ✓ | ✗ | ✗ | 185($\pm$11) | 9.18($\pm$0.13) | 94.94($\pm$1.32) |
| 2048 | 96 | 158.3 | ✓ | ✓ | ✗ | 152($\pm$7) | 8.73($\pm$0.45) | 98.76($\pm$2.84) |
| 2048 | 96 | 158.3 | ✓ | ✓ | ✓ | 165($\pm$13) | 8.51($\pm$0.32) | 99.31($\pm$2.10) |
| 2048 | 64 | 71.3 | ✓ | ✓ | ✓ | 371($\pm$7) | 10.48($\pm$0.10) | 86.90($\pm$0.61) |

Table 1: Fréchet Inception Distance (FID, lower is better) and Inception Score (IS, higher is better) for ablations of our proposed modifications. *Batch* is batch size, *Param* is total number of parameters, *Ch.* is the channel multiplier representing the number of units in each layer, *Shared* is using shared embeddings, *Skip-$z$* is using skip connections from the latent to multiple layers, *Ortho.* is Orthogonal Regularization, and *Itr* indicates if the setting is stable to $10^6$ iterations, or it collapses at the given iteration. Other than rows 1-4, results are computed across 8 random initializations.

Dumoulin et al. (2017) modify the way class conditioning is passed to **G** by supplying it with class-conditional gains and biases in BatchNorm (Ioffe & Szegedy, 2015) layers. In Miyato & Koyama (2018), **D** is conditioned by using the cosine similarity between its features and a set of learned class embeddings as additional evidence for distinguishing real and generated samples, effectively encouraging generation of samples whose features match a learned class prototype.

Objectively evaluating implicit generative models is difficult (Theis et al., 2015). A variety of works have proposed heuristics for measuring the sample quality of models without tractable likelihoods (Salimans et al., 2016; Heusel et al., 2017; Bińkowski et al., 2018; Wu et al., 2017). Of these, the Inception Score (IS, Salimans et al. (2016)) and Fréchet Inception Distance (FID, Heusel et al. (2017)) have become popular despite their notable flaws (Barratt & Sharma, 2018). We employ them as approximate measures of sample quality, and to enable comparison against previous work.

## 3 SCALING UP GANS

In this section, we explore methods for scaling up GAN training to reap the performance benefits of larger models and larger batches. As a baseline, we employ the SA-GAN architecture of Zhang et al. (2018), which uses the hinge loss (Lim & Ye, 2017; Tran et al., 2017) GAN objective. We provide class information to **G** with class-conditional BatchNorm (Dumoulin et al., 2017; de Vries et al., 2017) and to **D** with projection (Miyato & Koyama, 2018). The optimization settings follow Zhang et al. (2018) (notably employing Spectral Norm in **G**) with the modification that we halve the learning rates and take two **D** steps per **G** step. For evaluation, we employ moving averages of **G**'s weights following Karras et al. (2018); Mescheder et al. (2018); Yazıc et al. (2018), with a decay of 0.9999. We use Orthogonal Initialization (Saxe et al., 2014), whereas previous works used $\mathcal{N}(0, 0.02I)$ (Radford et al., 2016) or Xavier initialization (Glorot & Bengio, 2010). Each model is trained on 128 to 512 cores of a Google TPUv3 Pod (Google, 2018), and computes BatchNorm statistics in **G** across all devices, rather than per-device as is typical. We find progressive growing (Karras et al., 2018) unnecessary even for our $512 \times 512$ models. Additional details are in Appendix C.

We begin by increasing the batch size for the baseline model, and immediately find tremendous benefits in doing so. Rows 1-4 of Table 1 show that simply increasing the batch size by a factor of 8 improves the state-of-the-art IS by 46%. We conjecture that this is a result of each batch covering more modes, providing better gradients for both networks. One notable side effect of this scaling is that our models reach better final performance in fewer iterations, but become unstable and undergo complete training collapse. We discuss the causes and ramifications of this in Section 4. For these experiments, we report scores from checkpoints saved just before collapse.

We then increase the width (number of channels) in each layer by 50%, approximately doubling the number of parameters in both models. This leads to a further IS improvement of 21%, which we posit is due to the increased capacity of the model relative to the complexity of the dataset. Doubling

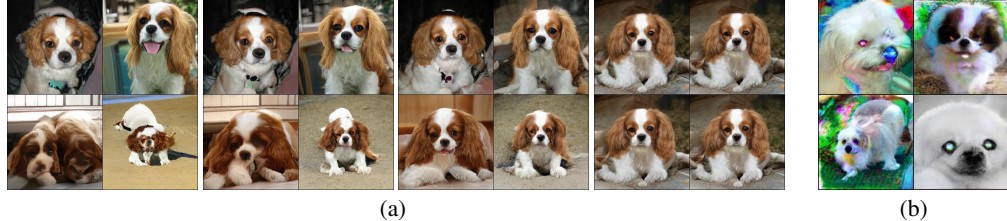

(a)                                                                                          (b)

Figure 2: (a) The effects of increasing truncation. From left to right, the threshold is set to 2, 1, 0.5, 0.04. (b) Saturation artifacts from applying truncation to a poorly conditioned model.

the depth did not initially lead to improvement – we addressed this later in the BigGAN-deep model, which uses a different residual block structure.

We note that class embeddings $c$ used for the conditional BatchNorm layers in **G** contain a large number of weights. Instead of having a separate layer for each embedding (Miyato et al., 2018; Zhang et al., 2018), we opt to use a shared embedding, which is linearly projected to each layer's gains and biases (Perez et al., 2018). This reduces computation and memory costs, and improves training speed (in number of iterations required to reach a given performance) by 37%. Next, we add direct skip connections (skip-$z$) from the noise vector $z$ to multiple layers of **G** rather than just the initial layer. The intuition behind this design is to allow **G** to use the latent space to directly influence features at different resolutions and levels of hierarchy. In BigGAN, this is accomplished by splitting $z$ into one chunk per resolution, and concatenating each chunk to the conditional vector $c$ which gets projected to the BatchNorm gains and biases. In BigGAN-deep, we use an even simpler design, concatenating the entire $z$ with the conditional vector without splitting it into chunks. Previous works (Goodfellow et al., 2014; Denton et al., 2015) have considered variants of this concept; our implementation is a minor modification of this design. Skip-$z$ provides a modest performance improvement of around 4%, and improves training speed by a further 18%.

### 3.1 TRADING OFF VARIETY AND FIDELITY WITH THE TRUNCATION TRICK

Unlike models which need to backpropagate through their latents, GANs can employ an arbitrary prior $p(z)$, yet the vast majority of previous works have chosen to draw $z$ from either $\mathcal{N}(0, I)$ or $\mathcal{U}[-1, 1]$. We question the optimality of this choice and explore alternatives in Appendix E.

Remarkably, our best results come from using a different latent distribution for sampling than was used in training. Taking a model trained with $z \sim \mathcal{N}(0, I)$ and sampling $z$ from a *truncated normal* (where values which fall outside a range are resampled to fall inside that range) immediately provides a boost to IS and FID. We call this the *Truncation Trick*: truncating a $z$ vector by resampling the values with magnitude above a chosen threshold leads to improvement in individual sample quality at the cost of reduction in overall sample variety. Figure 2(a) demonstrates this: as the threshold is reduced, and elements of $z$ are truncated towards zero (the mode of the latent distribution), individual samples approach the mode of **G**'s output distribution. Related observations about this trade-off were made in (Marchesi, 2016; Pieters & Wiering, 2014).

This technique allows fine-grained, post-hoc selection of the trade-off between sample quality and variety for a given **G**. Notably, we can compute FID and IS for a range of thresholds, obtaining the variety-fidelity curve reminiscent of the precision-recall curve (Figure 17). As IS does not penalize lack of variety in class-conditional models, reducing the truncation threshold leads to a direct increase in IS (analogous to precision). FID penalizes lack of variety (analogous to recall) but also rewards precision, so we initially see a moderate improvement in FID, but as truncation approaches zero and variety diminishes, the FID sharply drops. The distribution shift caused by sampling with different latents than those seen in training is problematic for many models. Some of our larger models are not amenable to truncation, producing saturation artifacts (Figure 2(b)) when fed truncated noise. To counteract this, we seek to enforce amenability to truncation by conditioning **G** to be smooth, so that the full space of $z$ will map to good output samples. For this, we turn to Orthogonal Regularization (Brock et al., 2017), which directly enforces the orthogonality condition:

$$R_\beta(W) = \beta\|W^\top W - I\|_{\mathrm{F}}^2, \tag{2}$$

where $W$ is a weight matrix and $\beta$ a hyperparameter. This regularization is known to often be too limiting (Miyato et al., 2018), so we explore several variants designed to relax the constraint while still imparting the desired smoothness to our models. The version we find to work best removes the diagonal terms from the regularization, and aims to minimize the pairwise cosine similarity between filters but does not constrain their norm:

$$R_\beta(W) = \beta\|W^\top W \odot (\mathbf{1} - I)\|_{\mathrm{F}}^2, \tag{3}$$

where $\mathbf{1}$ denotes a matrix with all elements set to 1. We sweep $\beta$ values and select $10^{-4}$, finding this small added penalty sufficient to improve the likelihood that our models will be amenable to truncation. Across runs in Table 1, we observe that without Orthogonal Regularization, only 16% of models are amenable to truncation, compared to 60% when trained with Orthogonal Regularization.

## 3.2 Summary

We find that current GAN techniques are sufficient to enable scaling to large models and distributed, large-batch training. We find that we can dramatically improve the state of the art and train models up to 512×512 resolution without need for explicit multiscale methods like Karras et al. (2018). Despite these improvements, our models undergo training collapse, necessitating early stopping in practice. In the next two sections we investigate why settings which were stable in previous works become unstable when applied at scale.

## 4 Analysis

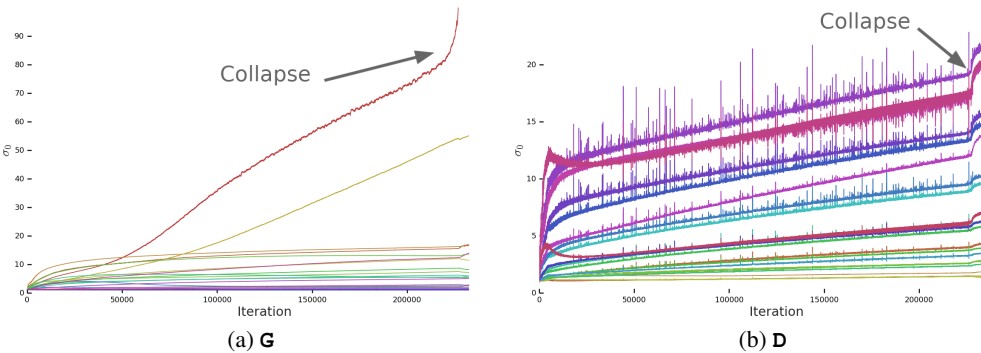

(a) **G**    (b) **D**

Figure 3: A typical plot of the first singular value $\sigma_0$ in the layers of **G** (a) and **D** (b) before Spectral Normalization. Most layers in **G** have well-behaved spectra, but without constraints a small subset grow throughout training and explode at collapse. **D**'s spectra are noisier but otherwise better-behaved. Colors from red to violet indicate increasing depth.

## 4.1 Characterizing Instability: The Generator

Much previous work has investigated GAN stability from a variety of analytical angles and on toy problems, but the instabilities we observe occur for settings which are stable at small scale, necessitating direct analysis at large scale. We monitor a range of weight, gradient, and loss statistics during training, in search of a metric which might presage the onset of training collapse, similar to (Odena et al., 2018). We found the top three singular values $\sigma_0, \sigma_1, \sigma_2$ of each weight matrix to be the most informative. They can be efficiently computed using the Alrnoldi iteration method (Golub & der Vorst, 2000), which extends the power iteration method, used in Miyato et al. (2018), to estimation of additional singular vectors and values. A clear pattern emerges, as can be seen in Figure 3(a) and Appendix F: most **G** layers have well-behaved spectral norms, but some layers

(typically the first layer in G, which is over-complete and not convolutional) are ill-behaved, with spectral norms that grow throughout training and explode at collapse.

To ascertain if this pathology is a cause of collapse or merely a symptom, we study the effects of imposing additional conditioning on G to explicitly counteract spectral explosion. First, we directly regularize the top singular values $\sigma_0$ of each weight, either towards a fixed value $\sigma_{reg}$ or towards some ratio $r$ of the second singular value, $r \cdot sg(\sigma_1)$ (with $sg$ the stop-gradient operation to prevent the regularization from increasing $\sigma_1$). Alternatively, we employ a partial singular value decomposition to instead clamp $\sigma_0$. Given a weight $W$, its first singular vectors $u_0$ and $v_0$, and $\sigma_{clamp}$ the value to which the $\sigma_0$ will be clamped, our weights become:

$$W = W - \max(0, \sigma_0 - \sigma_{clamp})v_0 u_0^\top, \tag{4}$$

where $\sigma_{clamp}$ is set to either $\sigma_{reg}$ or $r \cdot sg(\sigma_1)$. We observe that both with and without Spectral Normalization these techniques have the effect of preventing the gradual increase and explosion of either $\sigma_0$ or $\frac{\sigma_0}{\sigma_1}$, but even though in some cases they mildly improve performance, no combination prevents training collapse. This evidence suggests that while conditioning G might improve stability, it is insufficient to ensure stability. We accordingly turn our attention to D.

## 4.2 Characterizing Instability: The Discriminator

As with G, we analyze the spectra of D's weights to gain insight into its behavior, then seek to stabilize training by imposing additional constraints. Figure 3(b) displays a typical plot of $\sigma_0$ for D (with further plots in Appendix F). Unlike G, we see that the spectra are noisy, $\frac{\sigma_0}{\sigma_1}$ is well-behaved, and the singular values grow throughout training but only jump at collapse, instead of exploding.

The spikes in D's spectra might suggest that it periodically receives very large gradients, but we observe that the Frobenius norms are smooth (Appendix F), suggesting that this effect is primarily concentrated on the top few singular directions. We posit that this noise is a result of optimization through the adversarial training process, where G periodically produces batches which strongly perturb D . If this spectral noise is causally related to instability, a natural counter is to employ gradient penalties, which explicitly regularize changes in D's Jacobian. We explore the $R_1$ zero-centered gradient penalty from Mescheder et al. (2018):

$$R_1 := \frac{\gamma}{2}\mathbb{E}_{p_{\mathcal{D}}(x)}\left[\|\nabla D(x)\|_F^2\right]. \tag{5}$$

With the default suggested $\gamma$ strength of 10, training becomes stable and improves the smoothness and boundedness of spectra in both G and D, but performance severely degrades, resulting in a 45% reduction in IS. Reducing the penalty partially alleviates this degradation, but results in increasingly ill-behaved spectra; even with the penalty strength reduced to 1 (the lowest strength for which sudden collapse does not occur) the IS is reduced by 20%. Repeating this experiment with various strengths of Orthogonal Regularization, DropOut (Srivastava et al., 2014), and L2 (See Appendix I for details), reveals similar behaviors for these regularization strategies: with high enough penalties on D, training stability can be achieved, but at a substantial cost to performance.

We also observe that D's loss approaches zero during training, but undergoes a sharp upward jump at collapse (Appendix F). One possible explanation for this behavior is that D is overfitting to the training set, memorizing training examples rather than learning some meaningful boundary between real and generated images. As a simple test for D's memorization (related to Gulrajani et al. (2017)), we evaluate uncollapsed discriminators on the ImageNet training and validation sets, and measure what percentage of samples are classified as real or generated. While the training accuracy is consistently above 98%, the validation accuracy falls in the range of 50-55%, no better than random guessing (regardless of regularization strategy). This confirms that D is indeed memorizing the training set; we deem this in line with D's role, which is not explicitly to generalize, but to distill the training data and provide a useful learning signal for G. Additional experiments and discussion are provided in Appendix G.

## 4.3 Summary

We find that stability does not come solely from G or D, but from their interaction through the adversarial training process. While the symptoms of their poor conditioning can be used to track and

| Model | Res. | FID/IS | (min FID) / IS | FID / (valid IS) | FID / (max IS) |
|---|---|---|---|---|---|
| SN-GAN | 128 | 27.62/36.80 | N/A | N/A | N/A |
| SA-GAN | 128 | 18.65/52.52 | N/A | N/A | N/A |
| BigGAN | 128 | $8.7 \pm .6/98.8 \pm 3$ | $7.7 \pm .2/126.5 \pm 0$ | $9.6 \pm .4/166.3 \pm 1$ | $25 \pm 2/206 \pm 2$ |
| BigGAN | 256 | $8.7 \pm .1/142.3 \pm 2$ | $7.7 \pm .1/178.0 \pm 5$ | $9.3 \pm .3/233.1 \pm 1$ | $25 \pm 5/291 \pm 4$ |
| BigGAN | 512 | 8.1/144.2 | 7.6/170.3 | 11.8/241.4 | 27.0/275 |
| BigGAN-deep | 128 | $5.7 \pm .3/124.5 \pm 2$ | $6.3 \pm .3/148.1 \pm 4$ | $7.4 \pm .6/166.5 \pm 1$ | $25 \pm 2/253 \pm 11$ |
| BigGAN-deep | 256 | $6.9 \pm .2/171.4 \pm 2$ | $7.0 \pm .1/202.6 \pm 2$ | $8.1 \pm .1/232.5 \pm 2$ | $27 \pm 8/317 \pm 6$ |
| BigGAN-deep | 512 | 7.5/152.8 | 7.7/181.4 | 11.5/241.5 | 39.7/298 |

Table 2: Evaluation of models at different resolutions. We report scores without truncation (Column 3), scores at the best FID (Column 4), scores at the IS of validation data (Column 5), and scores at the max IS (Column 6). Standard deviations are computed over at least three random initializations.

identify instability, ensuring reasonable conditioning proves necessary for training but insufficient to prevent eventual training collapse. It is possible to enforce stability by strongly constraining D, but doing so incurs a dramatic cost in performance. With current techniques, better final performance can be achieved by relaxing this conditioning and allowing collapse to occur at the later stages of training, by which time a model is sufficiently trained to achieve good results.

## 5 EXPERIMENTS

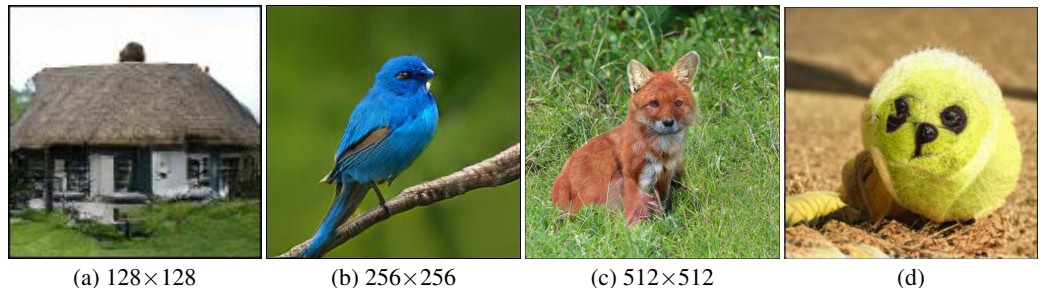

(a) 128×128          (b) 256×256          (c) 512×512          (d)

Figure 4: Samples from our BigGAN model with truncation threshold 0.5 (a-c) and an example of class leakage in a partially trained model (d).

### 5.1 EVALUATION ON IMAGENET

We evaluate our models on ImageNet ILSVRC 2012 (Russakovsky et al., 2015) at 128×128, 256×256, and 512×512 resolutions, employing the settings from Table 1, row 8. The samples generated by our models are presented in Figure 4, with additional samples in Appendix A, and online [2]. We report IS and FID in Table 2. As our models are able to trade sample variety for quality, it is unclear how best to compare against prior art; we accordingly report values at three settings, with complete curves in Appendix D. First, we report the FID/IS values at the truncation setting which attains the best FID. Second, we report the FID at the truncation setting for which our model's IS is the same as that attained by the real validation data, reasoning that this is a passable measure of maximum sample variety achieved while still achieving a good level of "objectness." Third, we report FID at the maximum IS achieved by each model, to demonstrate how much variety must be traded off to maximize quality. In all three cases, our models outperform the previous state-of-the-art IS and FID scores achieved by Miyato et al. (2018) and Zhang et al. (2018).

In addition to the BigGAN model introduced in the first version of the paper and used in the majority of experiments (unless otherwise stated), we also present a 4x deeper model (BigGAN-deep) which uses a different configuration of residual blocks. As can be seen from Table 2, BigGAN-deep substantially outperforms BigGAN across all resolutions and metrics. This confirms that our findings

---
[2]https://drive.google.com/drive/folders/1lWC6XEPD0LT5KUnPXeve_kWeY-FxH002

| Ch. | Param (M) | Shared | Skip-$z$ | Ortho. | FID | IS | (min FID) / IS | FID / (max IS) |
|---|---|---|---|---|---|---|---|---|
| 64 | 317.1 | ✗ | ✗ | ✗ | 48.38 | 23.27 | 48.6/23.1 | 49.1/23.9 |
| 64 | 99.4 | ✓ | ✓ | ✓ | 23.48 | 24.78 | 22.4/21.0 | 60.9/35.8 |
| 96 | 207.9 | ✓ | ✓ | ✓ | 18.84 | 27.86 | 17.1/23.3 | 51.6/38.1 |
| 128 | 355.7 | ✓ | ✓ | ✓ | 13.75 | 30.61 | 13.0/28.0 | 46.2/47.8 |

Table 3: BigGAN results on JFT-300M at $256\times256$ resolution. The *FID* and *IS* columns report these scores given by the JFT-300M-trained Inception v2 classifier with noise distributed as $z \sim \mathcal{N}(0, I)$ (non-truncated). The *(min FID) / IS* and *FID / (max IS)* columns report scores at the best FID and IS from a sweep across truncated noise distributions ranging from $\sigma = 0$ to $\sigma = 2$. Images from the JFT-300M validation set have an IS of 50.88 and FID of 1.94.

extend to other architectures, and that increased depth leads to improvement in sample quality. Both BigGAN and BigGAN-deep architectures are described in Appendix B.

Our observation that **D** overfits to the training set, coupled with our model's sample quality, raises the obvious question of whether or not **G** simply memorizes training points. To test this, we perform class-wise nearest neighbors analysis in pixel space and the feature space of pre-trained classifier networks (Appendix A). In addition, we present both interpolations between samples and class-wise interpolations (where $z$ is held constant) in Figures 8 and 9. Our model convincingly interpolates between disparate samples, and the nearest neighbors for its samples are visually distinct, suggesting that our model does not simply memorize training data.

We note that some failure modes of our partially-trained models are distinct from those previously observed. Most previous failures involve local artifacts (Odena et al., 2016), images consisting of texture blobs instead of objects (Salimans et al., 2016), or the canonical mode collapse. We observe *class leakage*, where images from one class contain properties of another, as exemplified by Figure 4(d). We also find that many classes on ImageNet are more difficult than others for our model; our model is more successful at generating dogs (which make up a large portion of the dataset, and are mostly distinguished by their texture) than crowds (which comprise a small portion of the dataset and have more large-scale structure). Further discussion is available in Appendix A.

## 5.2 Additional evaluation on JFT-300M

To confirm that our design choices are effective for even larger and more complex and diverse datasets, we also present results of our system on a subset of JFT-300M (Sun et al., 2017). The full JFT-300M dataset contains 300M real-world images labeled with 18K categories. Since the category distribution is heavily long-tailed, we subsample the dataset to keep only images with the 8.5K most common labels. The resulting dataset contains 292M images – two orders of magnitude larger than ImageNet. For images with multiple labels, we sample a single label randomly and independently whenever an image is sampled. To compute IS and FID for the GANs trained on this dataset, we use an Inception v2 classifier (Szegedy et al., 2016) trained on this dataset. Quantitative results are presented in Table 3. All models are trained with batch size 2048. We compare an ablated version of our model – comparable to SA-GAN (Zhang et al., 2018) but with the larger batch size – against a "full" BigGAN model that makes uses of all of the techniques applied to obtain the best results on ImageNet (shared embedding, skip-$z$, and orthogonal regularization). Our results show that these techniques substantially improve performance even in the setting of this much larger dataset at the same model capacity (64 base channels). We further show that for a dataset of this scale, we see significant additional improvements from expanding the capacity of our models to 128 base channels, while for ImageNet GANs that additional capacity was not beneficial.

In Figure 19 (Appendix D), we present truncation plots for models trained on this dataset. Unlike for ImageNet, where truncation limits of $\sigma \approx 0$ tend to produce the highest fidelity scores, IS is typically maximized for our JFT-300M models when the truncation value $\sigma$ ranges from 0.5 to 1. We suspect that this is at least partially due to the intra-class variability of JFT-300M labels, as well as the relative complexity of the image distribution, which includes images with multiple objects at a variety of scales. Interestingly, unlike models trained on ImageNet, where training tends to collapse without heavy regularization (Section 4), the models trained on JFT-300M remain stable over many

hundreds of thousands of iterations. This suggests that moving beyond ImageNet to larger datasets may partially alleviate GAN stability issues.

The improvement over the baseline GAN model that we achieve on this dataset without changes to the underlying models or training and regularization techniques (beyond expanded capacity) demonstrates that our findings extend from ImageNet to datasets with scale and complexity thus far unprecedented for generative models of images.

## 6 CONCLUSION

We have demonstrated that Generative Adversarial Networks trained to model natural images of multiple categories highly benefit from scaling up, both in terms of fidelity and variety of the generated samples. As a result, our models set a new level of performance among ImageNet GAN models, improving on the state of the art by a large margin. We have also presented an analysis of the training behavior of large scale GANs, characterized their stability in terms of the singular values of their weights, and discussed the interplay between stability and performance.

ACKNOWLEDGMENTS

We would like to thank Kai Arulkumaran, Matthias Bauer, Peter Buchlovsky, Jeffrey Defauw, Sander Dieleman, Ian Goodfellow, Ariel Gordon, Karol Gregor, Dominik Grewe, Chris Jones, Jacob Menick, Augustus Odena, Suman Ravuri, Ali Razavi, Mihaela Rosca, and Jeff Stanway.

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

# APPENDIX A    ADDITIONAL SAMPLES, INTERPOLATIONS, AND NEAREST NEIGHBORS FROM IMAGENET MODELS

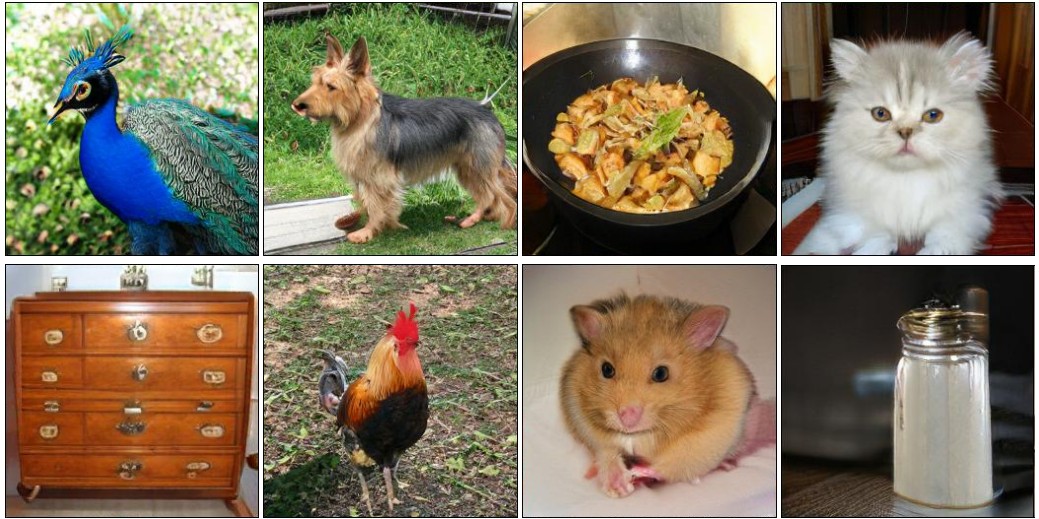

Figure 5: Samples generated by our BigGAN model at 256×256 resolution.

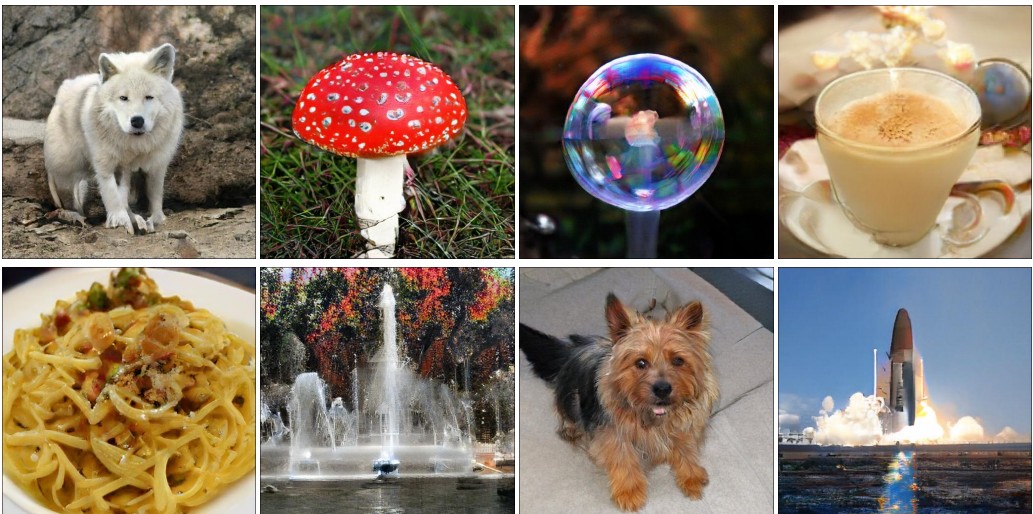

Figure 6: Samples generated by our BigGAN model at 512×512 resolution.

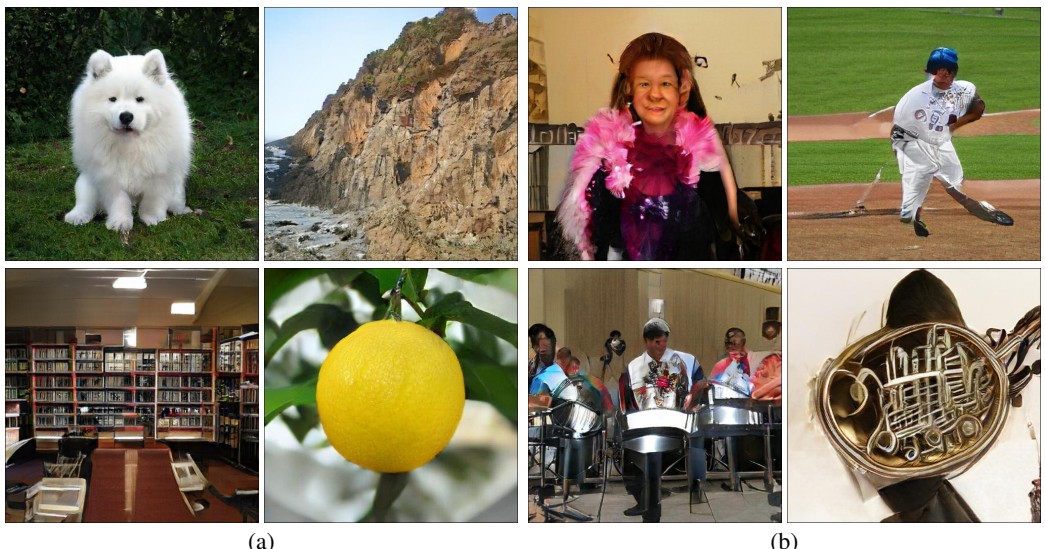

(a)                                    (b)

Figure 7: Comparing easy classes (a) with difficult classes (b) at 512×512. Classes such as dogs which are largely textural, and common in the dataset, are far easier to model than classes involving unaligned human faces or crowds. Such classes are more dynamic and structured, and often have details to which human observers are more sensitive. The difficulty of modeling global structure is further exacerbated when producing high-resolution images, even with non-local blocks.

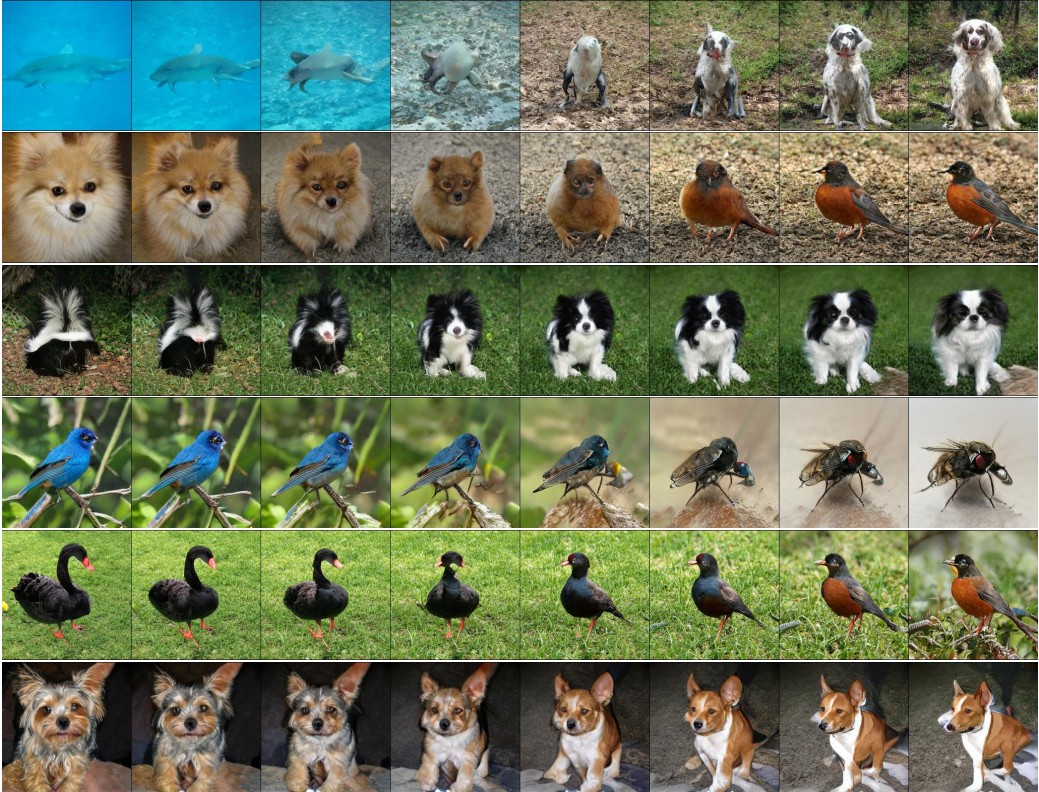

Figure 8: Interpolations between $z, c$ pairs.

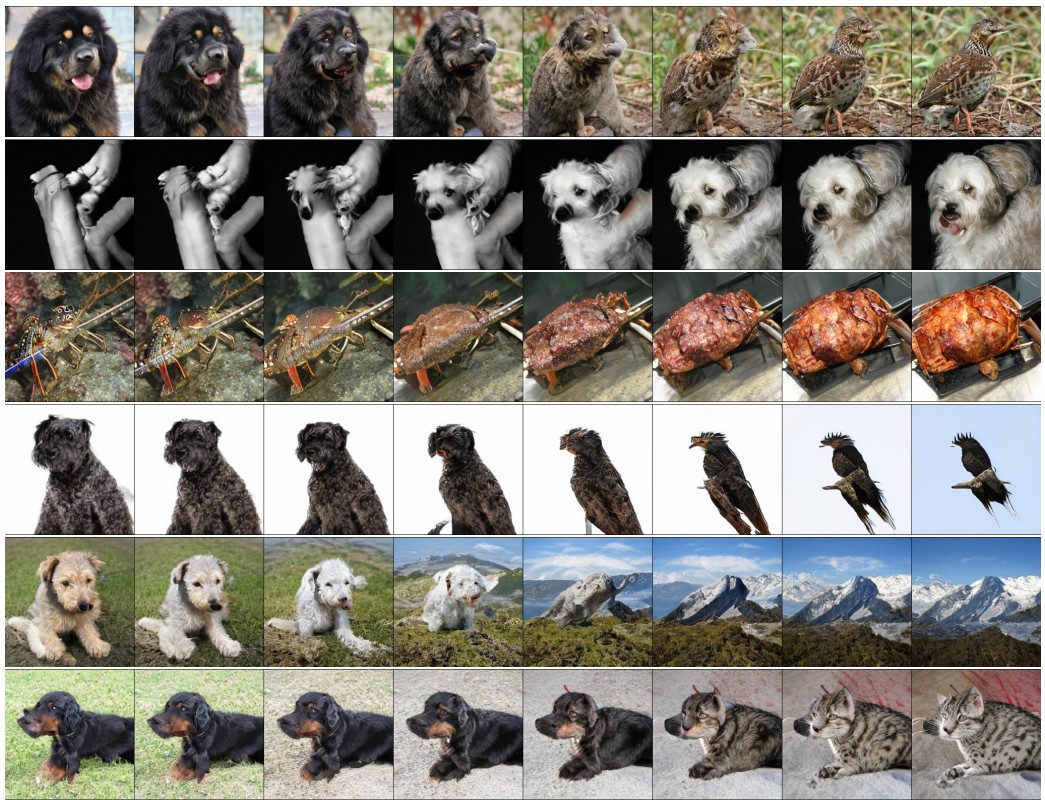

Figure 9: Interpolations between $c$ with $z$ held constant. Pose semantics are frequently maintained between endpoints (particularly in the final row). Row 2 demonstrates that grayscale is encoded in the joint $z, c$ space, rather than in $z$.

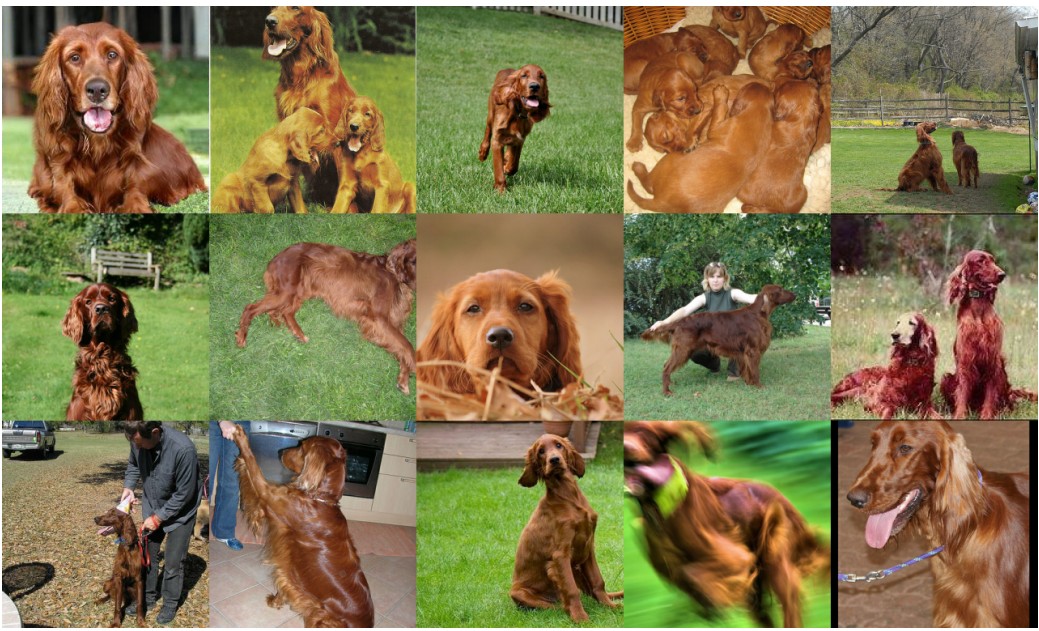

Figure 10: Nearest neighbors in VGG-16-fc7 (Simonyan & Zisserman, 2015) feature space. The generated image is in the top left.

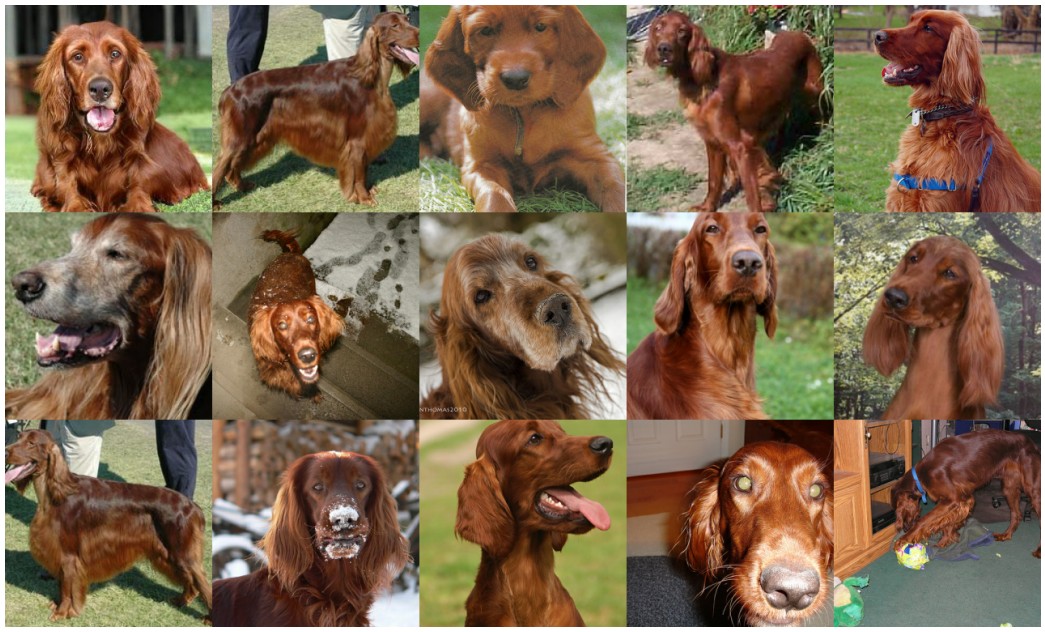

Figure 11: Nearest neighbors in ResNet-50-avgpool (He et al., 2016) feature space. The generated image is in the top left.

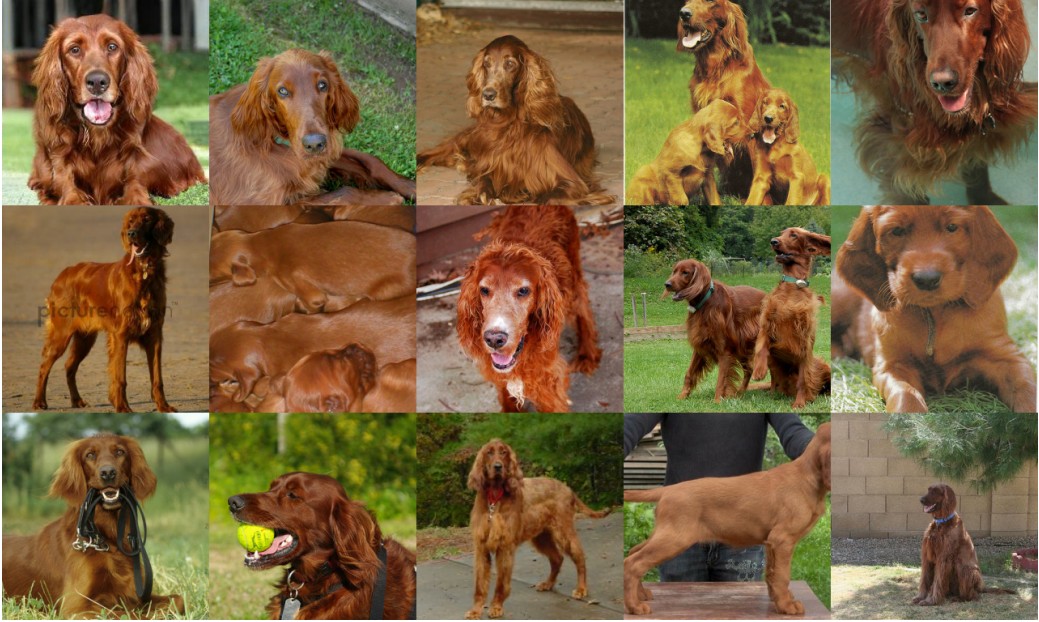

Figure 12: Nearest neighbors in pixel space. The generated image is in the top left.

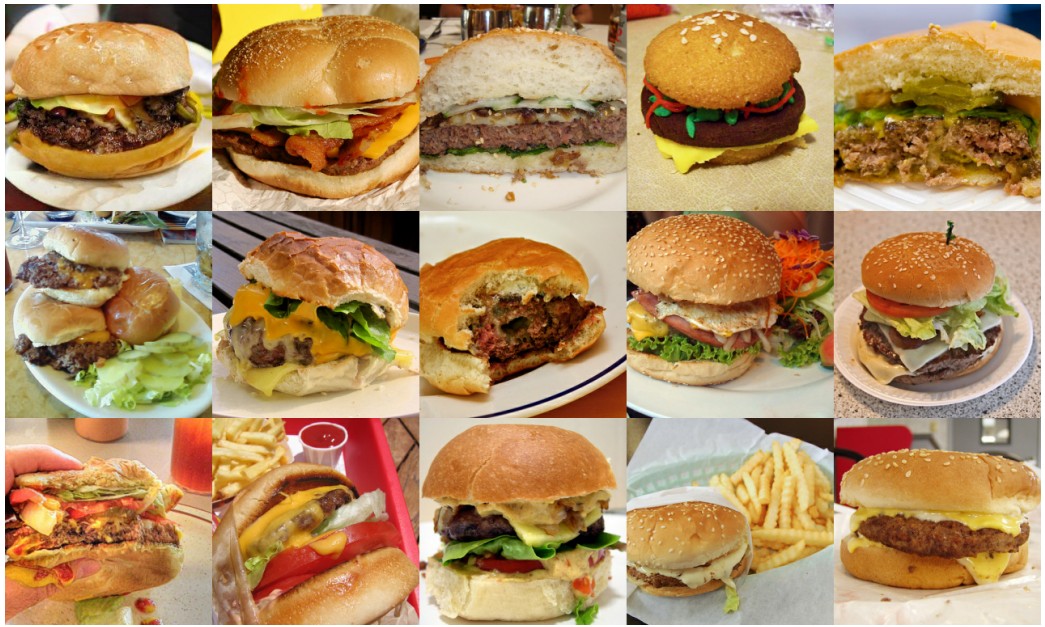

Figure 13: Nearest neighbors in VGG-16-fc7 (Simonyan & Zisserman, 2015) feature space. The generated image is in the top left.

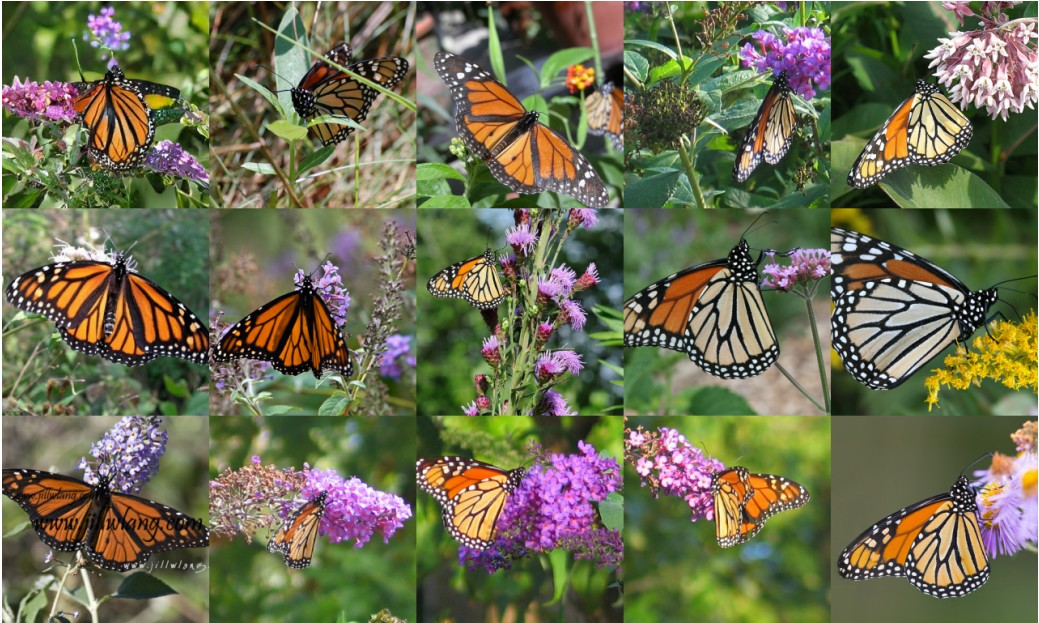

Figure 14: Nearest neighbors in ResNet-50-avgpool (He et al., 2016) feature space. The generated image is in the top left.

## APPENDIX B    ARCHITECTURAL DETAILS

In the BigGAN model (Figure 15), we use the ResNet (He et al., 2016) GAN architecture of (Zhang et al., 2018), which is identical to that used by (Miyato et al., 2018), but with the channel pattern in **D** modified so that the number of filters in the first convolutional layer of each block is equal to the number of output filters (rather than the number of input filters, as in Miyato et al. (2018); Gulrajani et al. (2017)). We use a single shared class embedding in **G**, and skip connections for the latent vector $z$ (skip-$z$). In particular, we employ hierarchical latent spaces, so that the latent vector $z$ is split along its channel dimension into chunks of equal size (20-D in our case), and each chunk is concatenated to the shared class embedding and passed to a corresponding residual block as a conditioning vector. The conditioning of each block is linearly projected to produce per-sample gains and biases for the BatchNorm layers of the block. The bias projections are zero-centered, while the gain projections are centered at 1. Since the number of residual blocks depends on the image resolution, the full dimensionality of $z$ is 120 for $128 \times 128$, 140 for $256 \times 256$, and 160 for $512 \times 512$ images.

The BigGAN-deep model (Figure 16) differs from BigGAN in several aspects. It uses a simpler variant of skip-$z$ conditioning: instead of first splitting $z$ into chunks, we concatenate the entire $z$ with the class embedding, and pass the resulting vector to each residual block through skip connections. BigGAN-deep is based on residual blocks with bottlenecks (He et al., 2016), which incorporate two additional $1 \times 1$ convolutions: the first reduces the number of channels by a factor of 4 before the more expensive $3 \times 3$ convolutions; the second produces the required number of output channels. While BigGAN relies on $1 \times 1$ convolutions in the skip connections whenever the number of channels needs to change, in BigGAN-deep we use a different strategy aimed at preserving identity throughout the skip connections. In **G**, where the number of channels needs to be reduced, we simply retain the first group of channels and drop the rest to produce the required number of channels. In **D**, where the number of channels should be increased, we pass the input channels unperturbed, and concatenate them with the remaining channels produced by a $1 \times 1$ convolution. As far as the network configuration is concerned, the discriminator is an exact reflection of the generator. There are two blocks at each resolution (BigGAN uses one), and as a result BigGAN-deep is four times deeper than BigGAN. Despite their increased depth, the BigGAN-deep models have significantly fewer parameters mainly due to the bottleneck structure of their residual blocks. For example, the $128 \times 128$ BigGAN-deep **G** and **D** have 50.4M and 34.6M parameters respectively, while the corresponding original BigGAN models have 70.4M and 88.0M parameters. All BigGAN-deep models use attention at $64 \times 64$ resolution, channel width multiplier $ch = 128$, and $z \in \mathbb{R}^{128}$.

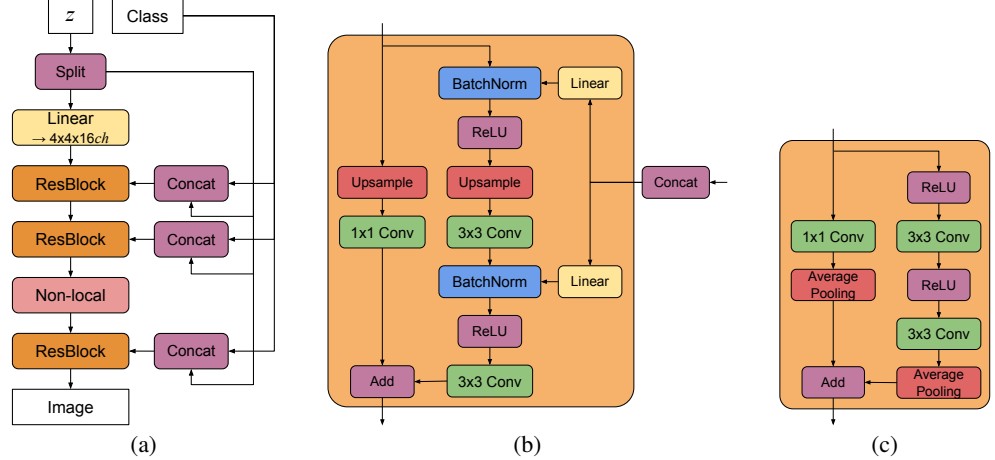

Figure 15: (a) A typical architectural layout for BigGAN's **G**; details are in the following tables. (b) A Residual Block (*ResBlock up*) in BigGAN's **G**. (c) A Residual Block (*ResBlock down*) in BigGAN's **D**.

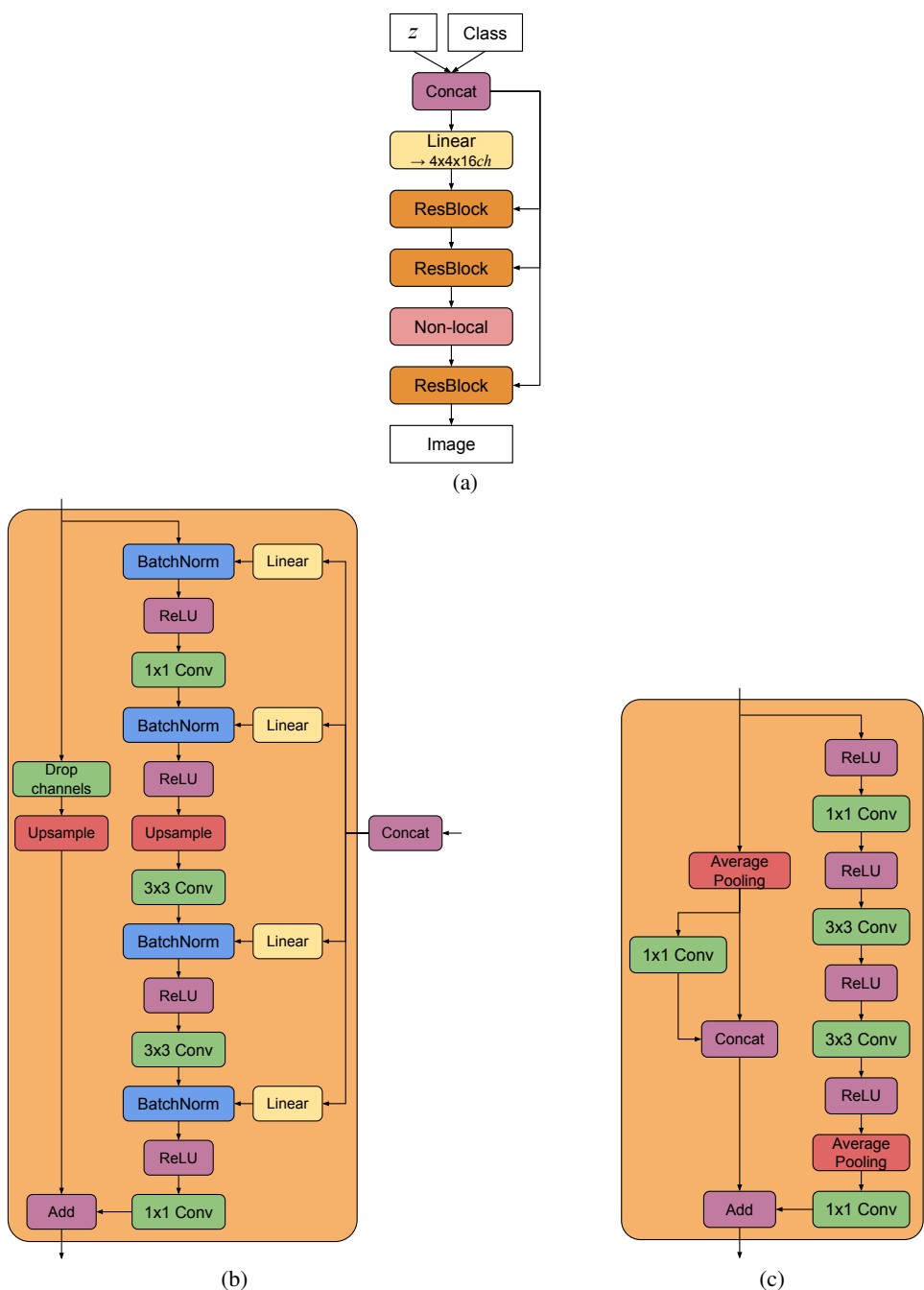

Figure 16: (a) A typical architectural layout for BigGAN-deep's **G**; details are in the following tables. (b) A Residual Block (*ResBlock up*) in BigGAN-deep's **G**. (c) A Residual Block (*ResBlock down*) in BigGAN-deep's **D**. A *ResBlock* (without *up* or *down*) in BigGAN-deep does not include the *Upsample* or *Average Pooling* layers, and has identity skip connections.

Table 4: BigGAN architecture for $128 \times 128$ images. $ch$ represents the channel width multiplier in each network from Table 1.

| $z \in \mathbb{R}^{120} \sim \mathcal{N}(0, I)$ |
| --- |
| Embed$(y) \in \mathbb{R}^{128}$ |
| Linear $(20 + 128) \rightarrow 4 \times 4 \times 16ch$ |
| ResBlock up $16ch \rightarrow 16ch$ |
| ResBlock up $16ch \rightarrow 8ch$ |
| ResBlock up $8ch \rightarrow 4ch$ |
| ResBlock up $4ch \rightarrow 2ch$ |
| Non-Local Block $(64 \times 64)$ |
| ResBlock up $2ch \rightarrow ch$ |
| BN, ReLU, $3 \times 3$ Conv $ch \rightarrow 3$ |
| Tanh |

(a) Generator

| RGB image $x \in \mathbb{R}^{128 \times 128 \times 3}$ |
| --- |
| ResBlock down $ch \rightarrow 2ch$ |
| Non-Local Block $(64 \times 64)$ |
| ResBlock down $2ch \rightarrow 4ch$ |
| ResBlock down $4ch \rightarrow 8ch$ |
| ResBlock down $8ch \rightarrow 16ch$ |
| ResBlock down $16ch \rightarrow 16ch$ |
| ResBlock $16ch \rightarrow 16ch$ |
| ReLU, Global sum pooling |
| Embed$(y) \cdot \boldsymbol{h}$ + (linear $\rightarrow 1$) |

(b) Discriminator

Table 5: BigGAN architecture for $256 \times 256$ images. Relative to the $128 \times 128$ architecture, we add an additional ResBlock in each network at $16 \times 16$ resolution, and move the non-local block in G to $128 \times 128$ resolution. Memory constraints prevent us from moving the non-local block in D.

| $z \in \mathbb{R}^{140} \sim \mathcal{N}(0, I)$ |
| --- |
| Embed$(y) \in \mathbb{R}^{128}$ |
| Linear $(20 + 128) \rightarrow 4 \times 4 \times 16ch$ |
| ResBlock up $16ch \rightarrow 16ch$ |
| ResBlock up $16ch \rightarrow 8ch$ |
| ResBlock up $8ch \rightarrow 8ch$ |
| ResBlock up $8ch \rightarrow 4ch$ |
| ResBlock up $4ch \rightarrow 2ch$ |
| Non-Local Block $(128 \times 128)$ |
| ResBlock up $2ch \rightarrow ch$ |
| BN, ReLU, $3 \times 3$ Conv $ch \rightarrow 3$ |
| Tanh |

(a) Generator

| RGB image $x \in \mathbb{R}^{256 \times 256 \times 3}$ |
| --- |
| ResBlock down $ch \rightarrow 2ch$ |
| ResBlock down $2ch \rightarrow 4ch$ |
| Non-Local Block $(64 \times 64)$ |
| ResBlock down $4ch \rightarrow 8ch$ |
| ResBlock down $8ch \rightarrow 8ch$ |
| ResBlock down $8ch \rightarrow 16ch$ |
| ResBlock down $16ch \rightarrow 16ch$ |
| ResBlock $16ch \rightarrow 16ch$ |
| ReLU, Global sum pooling |
| Embed$(y) \cdot \boldsymbol{h}$ + (linear $\rightarrow 1$) |

(b) Discriminator

Table 6: BigGAN architecture for $512 \times 512$ images. Relative to the $256 \times 256$ architecture, we add an additional ResBlock at the $512 \times 512$ resolution. Memory constraints force us to move the non-local block in both networks back to $64 \times 64$ resolution as in the $128 \times 128$ pixel setting.

| $z \in \mathbb{R}^{160} \sim \mathcal{N}(0, I)$ |
| :---: |
| Embed$(y) \in \mathbb{R}^{128}$ |
| Linear $(20 + 128) \to 4 \times 4 \times 16ch$ |
| ResBlock up $16ch \to 16ch$ |
| ResBlock up $16ch \to 8ch$ |
| ResBlock up $8ch \to 8ch$ |
| ResBlock up $8ch \to 4ch$ |
| Non-Local Block $(64 \times 64)$ |
| ResBlock up $4ch \to 2ch$ |
| ResBlock up $2ch \to ch$ |
| ResBlock up $ch \to ch$ |
| BN, ReLU, $3 \times 3$ Conv $ch \to 3$ |
| Tanh |

(a) Generator

| RGB image $x \in \mathbb{R}^{512 \times 512 \times 3}$ |
| :---: |
| ResBlock down $ch \to ch$ |
| ResBlock down $ch \to 2ch$ |
| ResBlock down $2ch \to 4ch$ |
| Non-Local Block $(64 \times 64)$ |
| ResBlock down $4ch \to 8ch$ |
| ResBlock down $8ch \to 8ch$ |
| ResBlock down $8ch \to 16ch$ |
| ResBlock down $16ch \to 16ch$ |
| ResBlock $16ch \to 16ch$ |
| ReLU, Global sum pooling |
| Embed$(y) \cdot \boldsymbol{h}$ + (linear $\to 1$) |

(b) Discriminator

Table 7: BigGAN-deep architecture for $128 \times 128$ images.

| $z \in \mathbb{R}^{128} \sim \mathcal{N}(0, I)$ |
| :---: |
| Embed$(y) \in \mathbb{R}^{128}$ |
| Linear $(128 + 128) \to 4 \times 4 \times 16ch$ |
| ResBlock $16ch \to 16ch$ |
| ResBlock up $16ch \to 16ch$ |
| ResBlock $16ch \to 16ch$ |
| ResBlock up $16ch \to 8ch$ |
| ResBlock $8ch \to 8ch$ |
| ResBlock up $8ch \to 4ch$ |
| ResBlock $4ch \to 4ch$ |
| ResBlock up $4ch \to 2ch$ |
| Non-Local Block $(64 \times 64)$ |
| ResBlock $2ch \to 2ch$ |
| ResBlock up $2ch \to ch$ |
| BN, ReLU, $3 \times 3$ Conv $ch \to 3$ |
| Tanh |

(a) Generator

| RGB image $x \in \mathbb{R}^{128 \times 128 \times 3}$ |
| :---: |
| $3 \times 3$ Conv $3 \to ch$ |
| ResBlock down $ch \to 2ch$ |
| ResBlock $2ch \to 2ch$ |
| Non-Local Block $(64 \times 64)$ |
| ResBlock down $2ch \to 4ch$ |
| ResBlock $4ch \to 4ch$ |
| ResBlock down $4ch \to 8ch$ |
| ResBlock $8ch \to 8ch$ |
| ResBlock down $8ch \to 16ch$ |
| ResBlock $16ch \to 16ch$ |
| ResBlock down $16ch \to 16ch$ |
| ResBlock $16ch \to 16ch$ |
| ReLU, Global sum pooling |
| Embed$(y) \cdot \boldsymbol{h}$ + (linear $\to 1$) |

(b) Discriminator

Table 8: BigGAN-deep architecture for $256 \times 256$ images.

| $z \in \mathbb{R}^{128} \sim \mathcal{N}(0, I)$ Embed$(y) \in \mathbb{R}^{128}$ |
| :---: |
| Linear $(128 + 128) \rightarrow 4 \times 4 \times 16ch$ |
| ResBlock $16ch \rightarrow 16ch$ |
| ResBlock up $16ch \rightarrow 16ch$ |
| ResBlock $16ch \rightarrow 16ch$ |
| ResBlock up $16ch \rightarrow 8ch$ |
| ResBlock $8ch \rightarrow 8ch$ |
| ResBlock up $8ch \rightarrow 8ch$ |
| ResBlock $8ch \rightarrow 8ch$ |
| ResBlock up $8ch \rightarrow 4ch$ |
| Non-Local Block $(64 \times 64)$ |
| ResBlock $4ch \rightarrow 4ch$ |
| ResBlock up $4ch \rightarrow 2ch$ |
| ResBlock $2ch \rightarrow 2ch$ |
| ResBlock up $2ch \rightarrow ch$ |
| BN, ReLU, $3 \times 3$ Conv $ch \rightarrow 3$ |
| Tanh |

(a) Generator

| RGB image $x \in \mathbb{R}^{256 \times 256 \times 3}$ |
| :---: |
| $3 \times 3$ Conv $3 \rightarrow ch$ |
| ResBlock down $ch \rightarrow 2ch$ |
| ResBlock $2ch \rightarrow 2ch$ |
| ResBlock down $2ch \rightarrow 4ch$ |
| ResBlock $4ch \rightarrow 4ch$ |
| Non-Local Block $(64 \times 64)$ |
| ResBlock down $4ch \rightarrow 8ch$ |
| ResBlock $8ch \rightarrow 8ch$ |
| ResBlock down $8ch \rightarrow 8ch$ |
| ResBlock $8ch \rightarrow 8ch$ |
| ResBlock down $8ch \rightarrow 16ch$ |
| ResBlock $16ch \rightarrow 16ch$ |
| ResBlock down $16ch \rightarrow 16ch$ |
| ResBlock $16ch \rightarrow 16ch$ |
| ReLU, Global sum pooling |
| Embed$(y) \cdot \boldsymbol{h}$ + (linear $\rightarrow 1$) |

(b) Discriminator

Table 9: BigGAN-deep architecture for $512 \times 512$ images.

| $z \in \mathbb{R}^{128} \sim \mathcal{N}(0, I)$ $\text{Embed}(y) \in \mathbb{R}^{128}$ |
|---|
| Linear $(128 + 128) \to 4 \times 4 \times 16ch$ |
| ResBlock $16ch \to 16ch$ |
| ResBlock up $16ch \to 16ch$ |
| ResBlock $16ch \to 16ch$ |
| ResBlock up $16ch \to 8ch$ |
| ResBlock $8ch \to 8ch$ |
| ResBlock up $8ch \to 8ch$ |
| ResBlock $8ch \to 8ch$ |
| ResBlock up $8ch \to 4ch$ |
| Non-Local Block $(64 \times 64)$ |
| ResBlock $4ch \to 4ch$ |
| ResBlock up $4ch \to 2ch$ |
| ResBlock $2ch \to 2ch$ |
| ResBlock up $2ch \to ch$ |
| ResBlock $ch \to ch$ |
| ResBlock up $ch \to ch$ |
| BN, ReLU, $3 \times 3$ Conv $ch \to 3$ |
| Tanh |

(a) Generator

| RGB image $x \in \mathbb{R}^{512 \times 512 \times 3}$ |
|---|
| $3 \times 3$ Conv $3 \to ch$ |
| ResBlock down $ch \to ch$ |
| ResBlock $ch \to ch$ |
| ResBlock down $ch \to 2ch$ |
| ResBlock $2ch \to 2ch$ |
| ResBlock down $2ch \to 4ch$ |
| ResBlock $4ch \to 4ch$ |
| Non-Local Block $(64 \times 64)$ |
| ResBlock down $4ch \to 8ch$ |
| ResBlock $8ch \to 8ch$ |
| ResBlock down $8ch \to 8ch$ |
| ResBlock $8ch \to 8ch$ |
| ResBlock down $8ch \to 16ch$ |
| ResBlock $16ch \to 16ch$ |
| ResBlock down $16ch \to 16ch$ |
| ResBlock $16ch \to 16ch$ |
| ReLU, Global sum pooling |
| $\text{Embed}(y) \cdot \boldsymbol{h} + (\text{linear} \to 1)$ |

(b) Discriminator

## Appendix C    Experimental Details

Our basic setup follows SA-GAN (Zhang et al., 2018), and is implemented in TensorFlow (Abadi et al., 2016). We employ the architectures detailed in Appendix B, with non-local blocks inserted at a single stage in each network. Both G and D networks are initialized with Orthogonal Initialization (Saxe et al., 2014). We use Adam optimizer (Kingma & Ba, 2014) with $\beta_1 = 0$ and $\beta_2 = 0.999$ and a constant learning rate. For BigGAN models at all resolutions, we use $2 \cdot 10^{-4}$ in D and $5 \cdot 10^{-5}$ in G. For BigGAN-deep, we use the learning rate of $2 \cdot 10^{-4}$ in D and $5 \cdot 10^{-5}$ in G for $128 \times 128$ models, and $2.5 \cdot 10^{-5}$ in both D and G for $256 \times 256$ and $512 \times 512$ models. We experimented with the number of D steps per G step (varying it from 1 to 6) and found that two D steps per G step gave the best results.

We use an exponential moving average of the weights of G at sampling time, with a decay rate set to 0.9999. We employ cross-replica BatchNorm (Ioffe & Szegedy, 2015) in G, where batch statistics are aggregated across all devices, rather than a single device as in standard implementations. Spectral Normalization (Miyato et al., 2018) is used in both G and D, following SA-GAN (Zhang et al., 2018). We train on a Google TPU v3 Pod, with the number of cores proportional to the resolution: 128 for $128 \times 128$, 256 for $256 \times 256$, and 512 for $512 \times 512$. Training takes between 24 and 48 hours for most models. We increase $\epsilon$ from the default $10^{-8}$ to $10^{-4}$ in BatchNorm and Spectral Norm to mollify low-precision numerical issues. We preprocess data by cropping along the long edge and rescaling to a given resolution with area resampling.

### C.1    BatchNorm Statistics and Sampling

The default behavior with batch normalized classifier networks is to use a running average of the activation moments at test time. Previous works (Radford et al., 2016) have instead used batch statistics when sampling images. While this is not technically an invalid way to sample, it means that results are dependent on the test batch size (and how many devices it is split across), and further complicates reproducibility.

We find that this detail is extremely important, with changes in test batch size producing drastic changes in performance. This is further exacerbated when one uses exponential moving averages of G's weights for sampling, as the BatchNorm running averages are computed with non-averaged weights and are poor estimates of the activation statistics for the averaged weights.

To counteract both these issues, we employ "standing statistics," where we compute activation statistics at sampling time by running the G through multiple forward passes (typically 100) each with different batches of random noise, and storing means and variances aggregated across all forward passes. Analogous to using running statistics, this results in G's outputs becoming invariant to batch size and the number of devices, even when producing a single sample.

### C.2    CIFAR-10

We run our networks on CIFAR-10 (Krizhevsky & Hinton, 2009) using the settings from Table 1, row 8, and achieve an IS of 9.22 and an FID of 14.73 without truncation.

### C.3    Inception Scores of ImageNet Images

We compute the IS for both the training and validation sets of ImageNet. At $128 \times 128$ the training data has an IS of 233, and the validation data has an IS of 166. At $256 \times 256$ the training data has an IS of 377, and the validation data has an IS of 234. At $512 \times 512$ the training data has an IS of 348, and the validation data has an IS of 241. The discrepancy between training and validation scores is due to the Inception classifier having been trained on the training data, resulting in high-confidence outputs that are preferred by the Inception Score.

## APPENDIX D    ADDITIONAL PLOTS

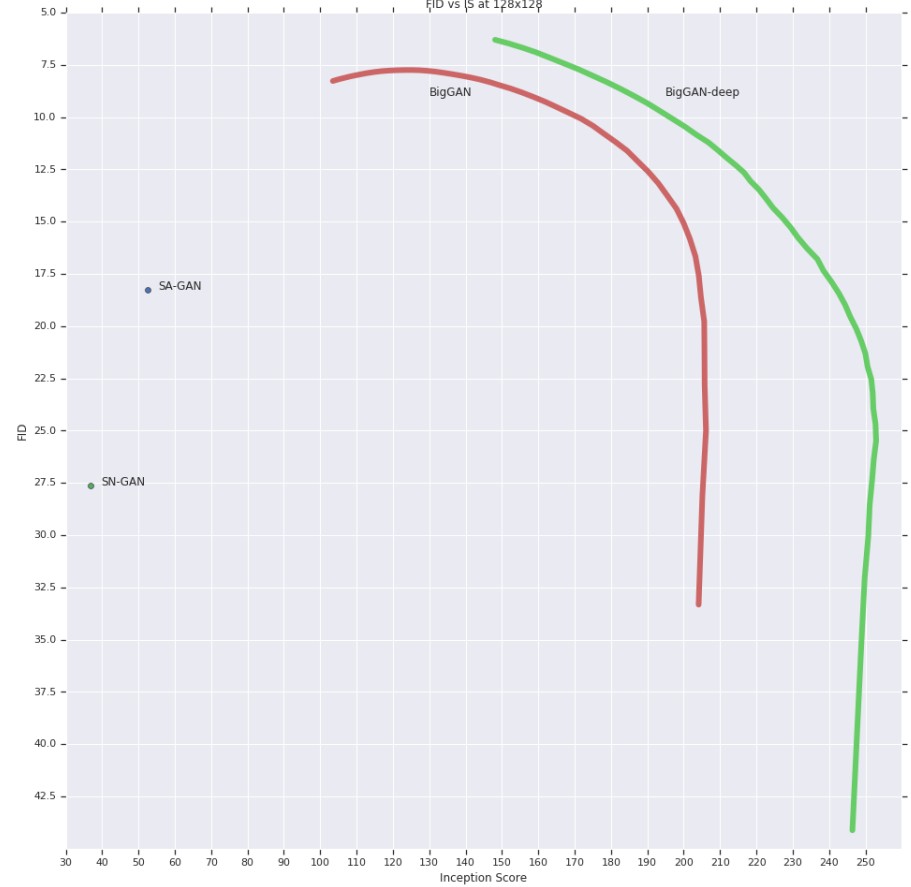

Figure 17: IS vs. FID at 128×128. Scores are averaged across three random seeds.

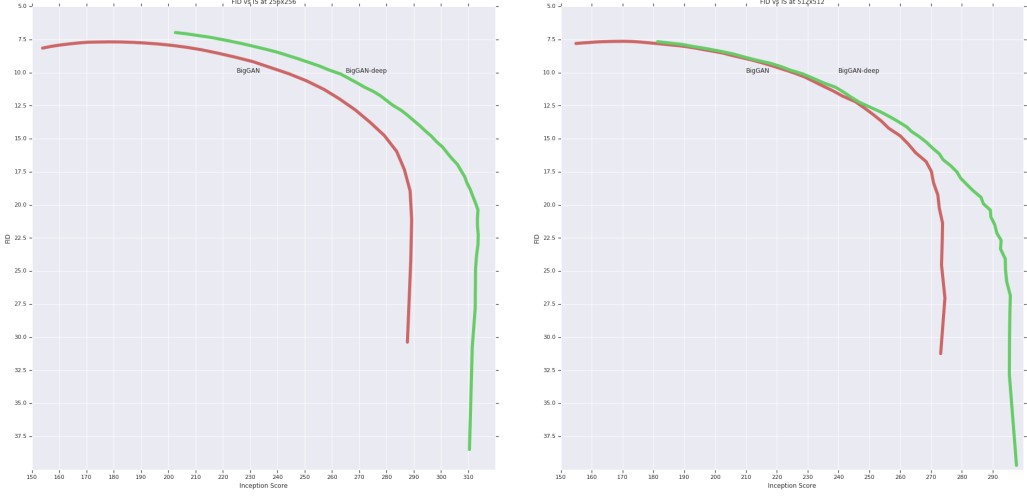

Figure 18: IS vs. FID at 256 and 512 pixels. Scores are averaged across three random seeds for 256.

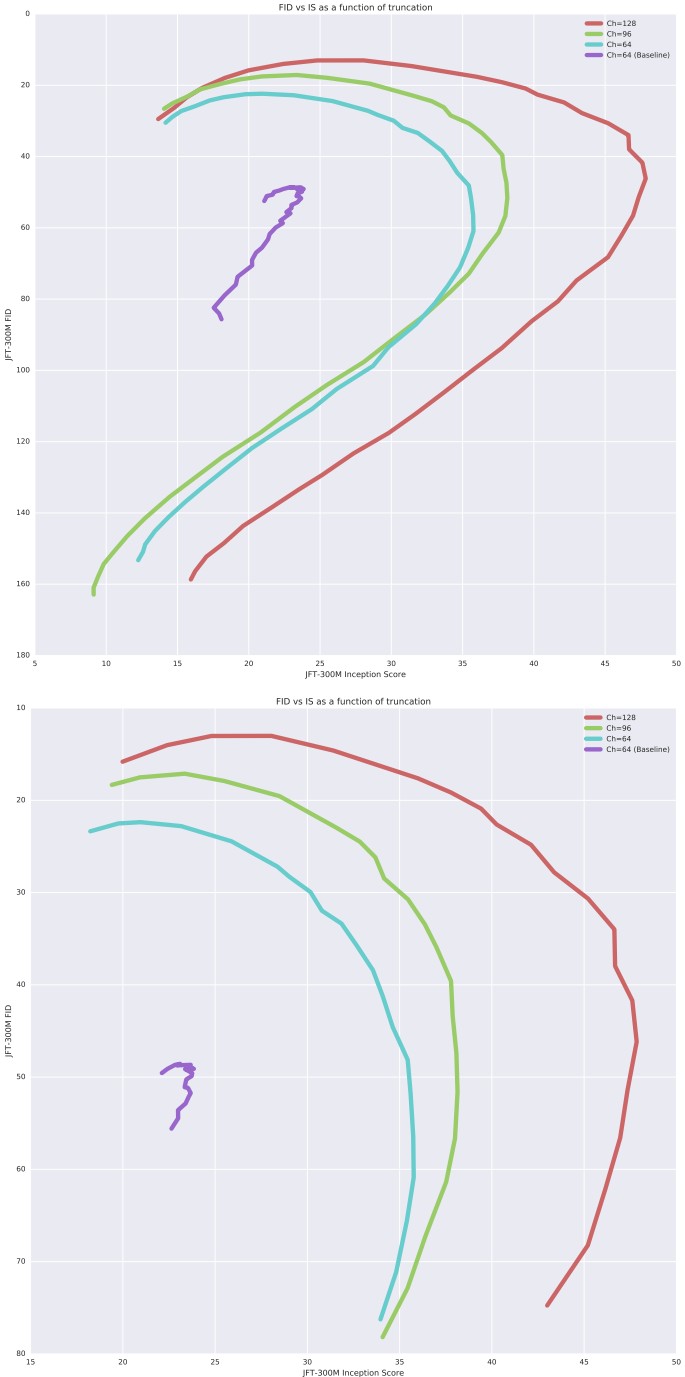

Figure 19: JFT-300M IS vs. FID at 256×256. We show truncation values from $\sigma = 0$ to $\sigma = 2$ (top) and from $\sigma = 0.5$ to $\sigma = 1.5$ (bottom). Each curve corresponds to a row in Table 3. The curve labeled with *baseline* corresponds to the first row (with orthogonal regularization and other techniques disabled), while the rest correspond to rows 2-4 – the same architecture at different capacities (*Ch*).

## APPENDIX E   CHOOSING LATENT SPACES

While most previous work has employed $\mathcal{N}(0, I)$ or $\mathcal{U}[-1, 1]$ as the prior for $z$ (the noise input to G), we are free to choose any latent distribution from which we can sample. We explore the choice of latents by considering an array of possible designs, described below. For each latent, we provide the intuition behind its design and briefly describe how it performs when used as a drop-in replacement for $z \sim \mathcal{N}(0, I)$ in an SA-GAN baseline. As the Truncation Trick proved more beneficial than switching to any of these latents, we do not perform a full ablation study, and employ $z \sim \mathcal{N}(0, I)$ for our main results to take full advantage of truncation. The two latents which we find to work best without truncation are Bernoulli $\{0, 1\}$ and Censored Normal $\max(\mathcal{N}(0, I), 0)$, both of which improve speed of training and lightly improve final performance, but are less amenable to truncation.

We also ablate the choice of latent space dimensonality (which by default is $z \in \mathbb{R}^{128}$), finding that we are able to successfully train with latent dimensions as low as $z \in \mathbb{R}^8$, and that with $z \in \mathbb{R}^{32}$ we see a minimal drop in performance. While this is substantially smaller than many previous works, direct comparison to single-class networks (such as those in Karras et al. (2018), which employ a $z \in \mathbb{R}^{512}$ latent space on a highly constrained dataset with 30,000 images) is improper, as our networks have additional class information provided as input.

### LATENTS

- $\mathcal{N}(0, I)$. A standard choice of the latent space which we use in the main experiments.

- $\mathcal{U}[-1, 1]$. Another standard choice; we find that it performs similarly to $\mathcal{N}(0, I)$.

- Bernoulli $\{0, 1\}$. A discrete latent might reflect our prior that underlying factors of variation in natural images are not continuous, but discrete (one feature is present, another is not). This latent outperforms $\mathcal{N}(0, I)$ (in terms of IS) by 8% and requires 60% fewer iterations.

- $\max(\mathcal{N}(0, I), 0)$, also called Censored Normal. This latent is designed to introduce sparsity in the latent space (reflecting our prior that certain latent features are sometimes present and sometimes not), but also allow those latents to vary continuously, expressing different degrees of intensity for latents which are active. This latent outperforms $\mathcal{N}(0, I)$ (in terms of IS) by 15-20% and tends to require fewer iterations.

- Bernoulli $\{-1, 1\}$. This latent is designed to be discrete, but not sparse (as the network can learn to activate in response to negative inputs). This latent performs near-identically to $\mathcal{N}(0, I)$.

- Independent Categorical in $\{-1, 0, 1\}$, with equal probability. This distribution is chosen to be discrete and have sparsity, but also to allow latents to take on both positive and negative values. This latent performs near-identically to $\mathcal{N}(0, I)$.

- $\mathcal{N}(0, I)$ multiplied by Bernoulli $\{0, 1\}$. This distribution is chosen to have continuous latent factors which are also sparse (with a peak at zero), similar to Censored Normal but not constrained to be positive. This latent performs near-identically to $\mathcal{N}(0, I)$.

- Concatenating $\mathcal{N}(0, I)$ and Bernoulli $\{0, 1\}$, each taking half of the latent dimensions. This is inspired by Chen et al. (2016), and is chosen to allow some factors of variation to be discrete, while others are continuous. This latent outperforms $\mathcal{N}(0, I)$ by around 5%.

- Variance annealing: we sample from $\mathcal{N}(0, \sigma I)$, where $\sigma$ is allowed to vary over training. We compared a variety of piecewise schedules and found that starting with $\sigma = 2$ and annealing towards $\sigma = 1$ over the course of training mildly improved performance. The space of possible variance schedules is large, and we did not explore it in depth – we suspect that a more principled or better-tuned schedule could more strongly impact performance.

- Per-sample variable variance: $\mathcal{N}(0, \sigma_i I)$, where $\sigma_i \sim \mathcal{U}[\sigma_l, \sigma_h]$ independently for each sample $i$ in a batch, and $(\sigma_l, \sigma_h)$ are hyperparameters. This distribution was chosen to try and improve amenability to the Truncation Trick by feeding the network noise samples with non-constant variance. This did not appear to affect performance, but we did not explore it in depth. One might also consider scheduling $(\sigma_l, \sigma_h)$, similar to variance annealing.

# APPENDIX F MONITORED TRAINING STATISTICS

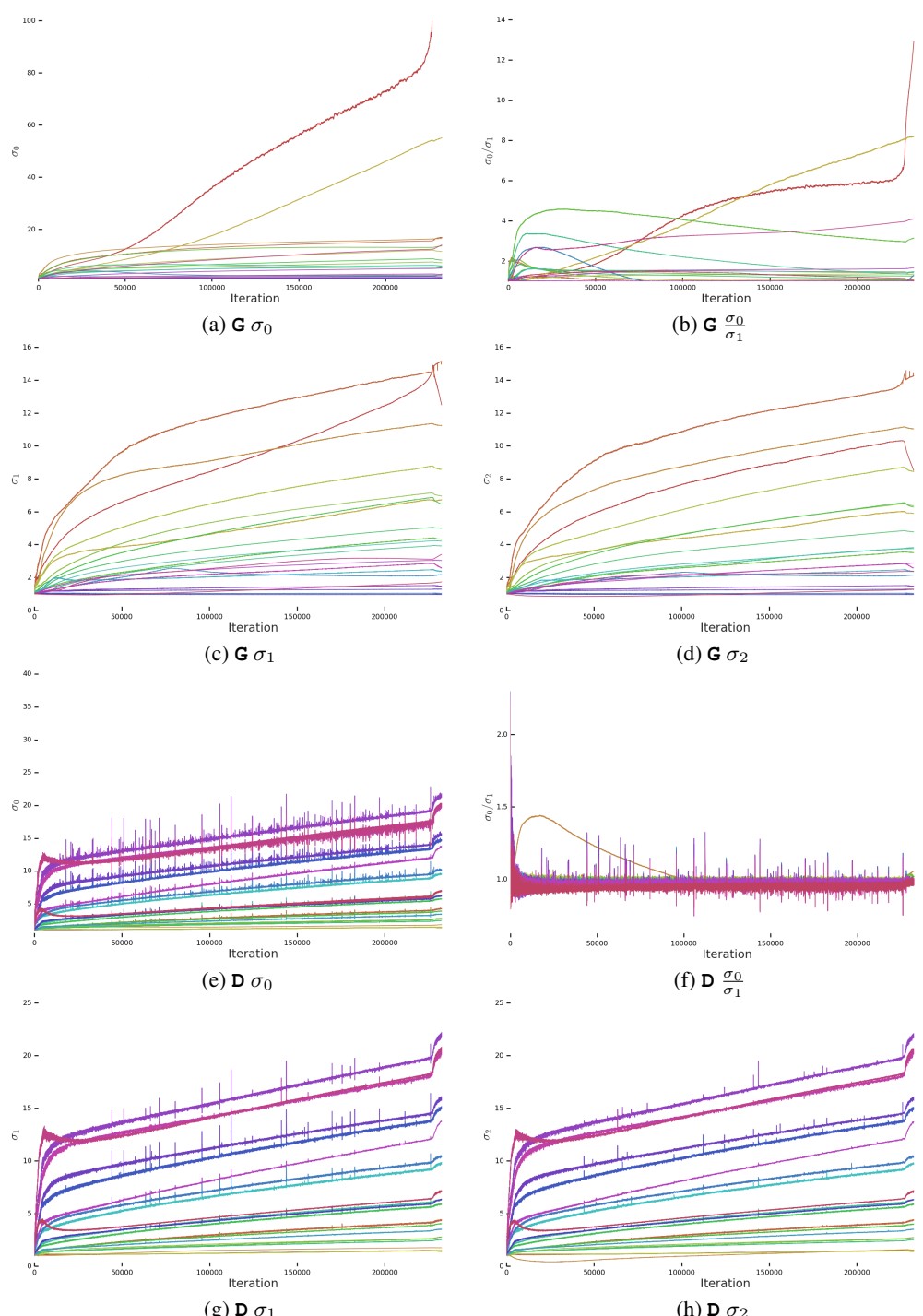

Figure 20: Training statistics for a typical model without special modifications. Collapse occurs after 200000 iterations.

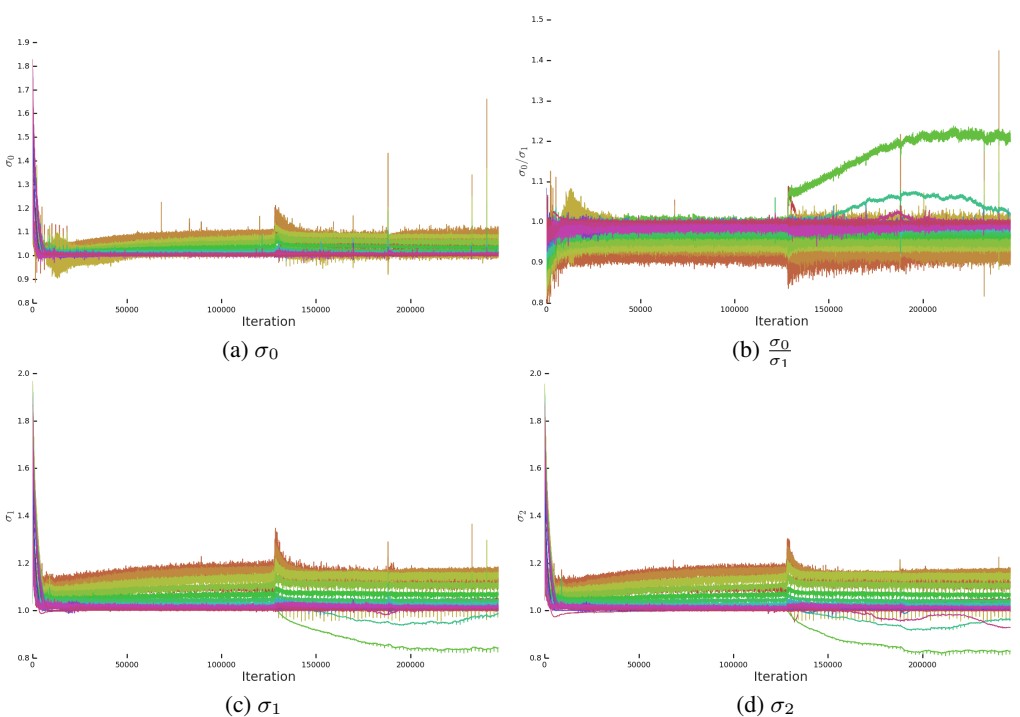

Figure 21: **G** training statistics with $\sigma_0$ in **G** regularized towards 1. Collapse occurs after 125000 iterations.

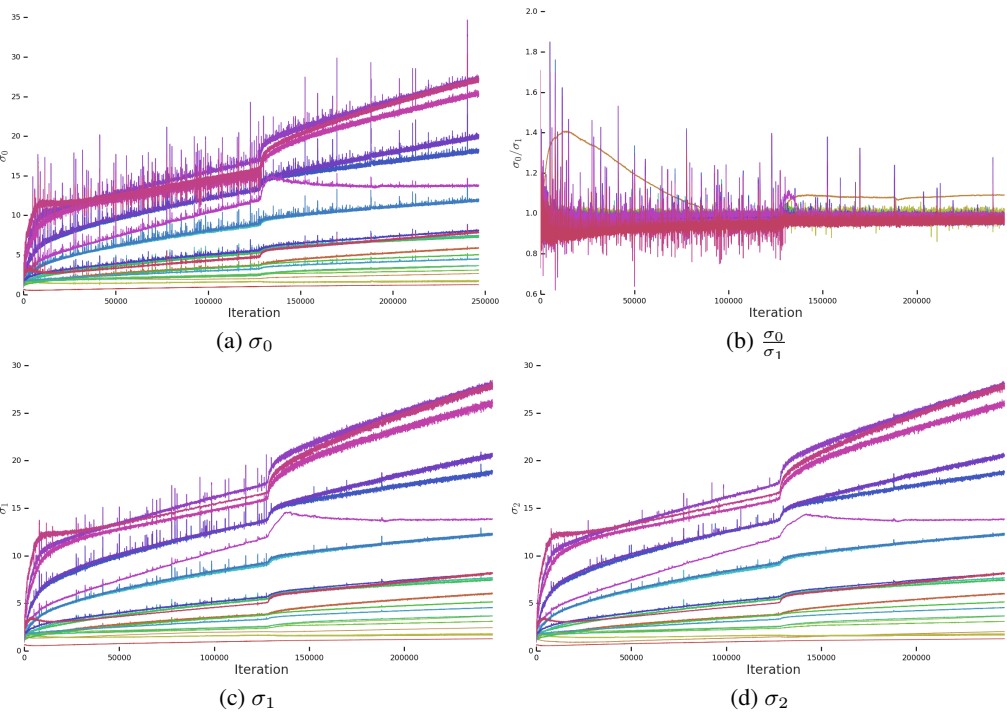

Figure 22: **D** training statistics with $\sigma_0$ in **G** regularized towards 1. Collapse occurs after 125000 iterations.

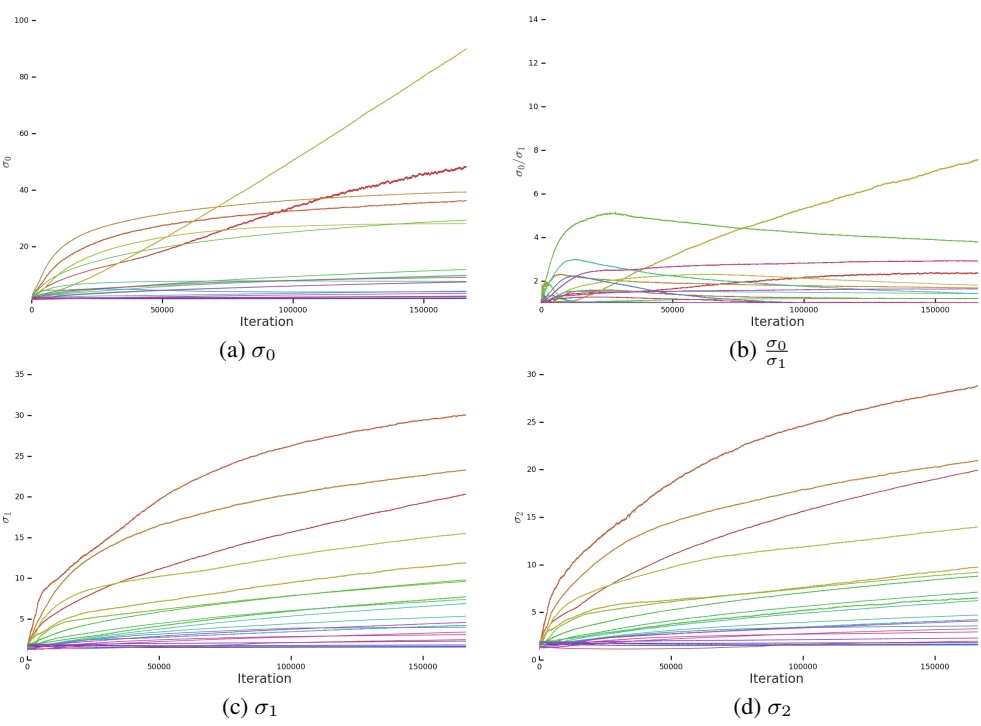

Figure 23: **G** training statistics with an R1 Gradient Penalty of strength 10 on **D**. This model does not collapse, but only reaches a maximum IS of 55.

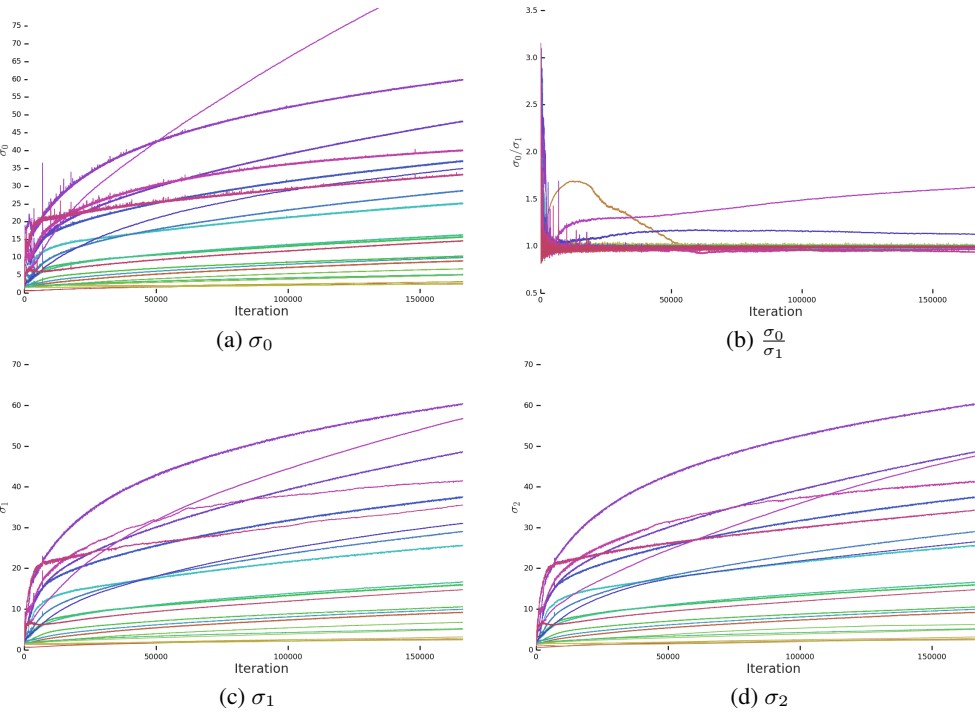

Figure 24: **D** training statistics with an R1 Gradient Penalty of strength 10 on **D**. This model does not collapse, but only reaches a maximum IS of 55.

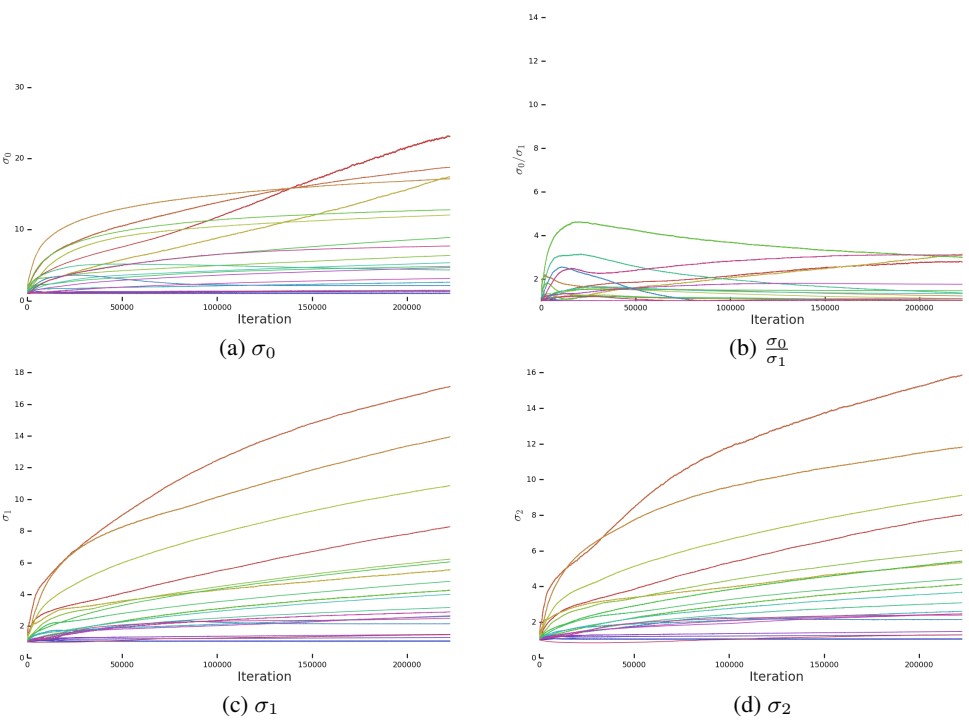

Figure 25: **G** training statistics with Dropout (keep probability 0.8) applied to the last feature layer of **D**. This model does not collapse, but only reaches a maximum IS of 70.

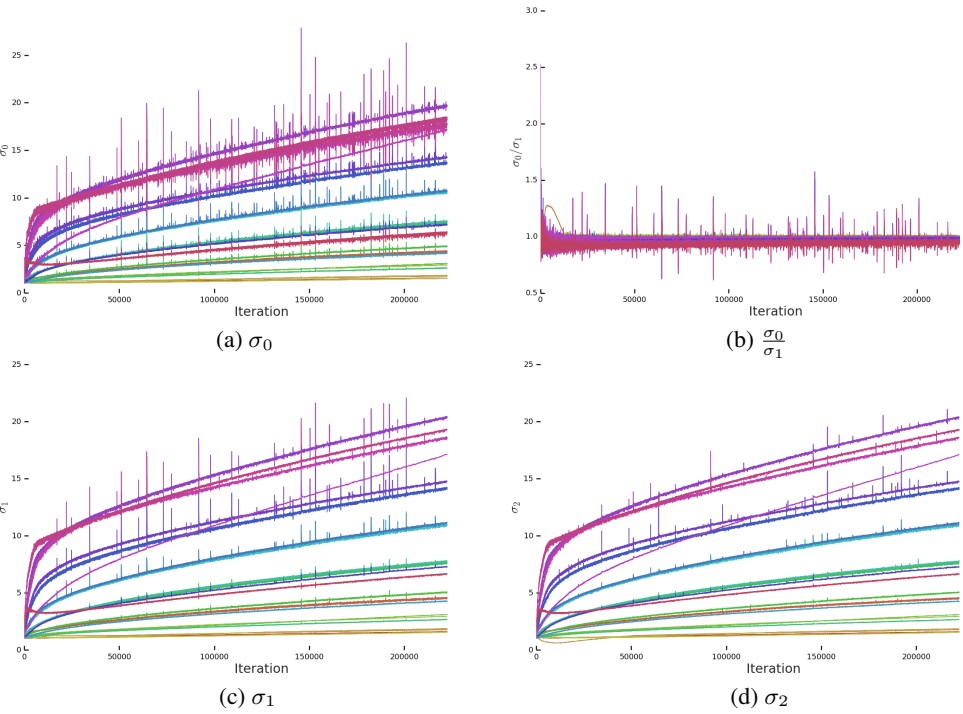

Figure 26: **D** training statistics with Dropout (keep probability 0.8) applied to the last feature layer of **D**. This model does not collapse, but only reaches a maximum IS of 70.

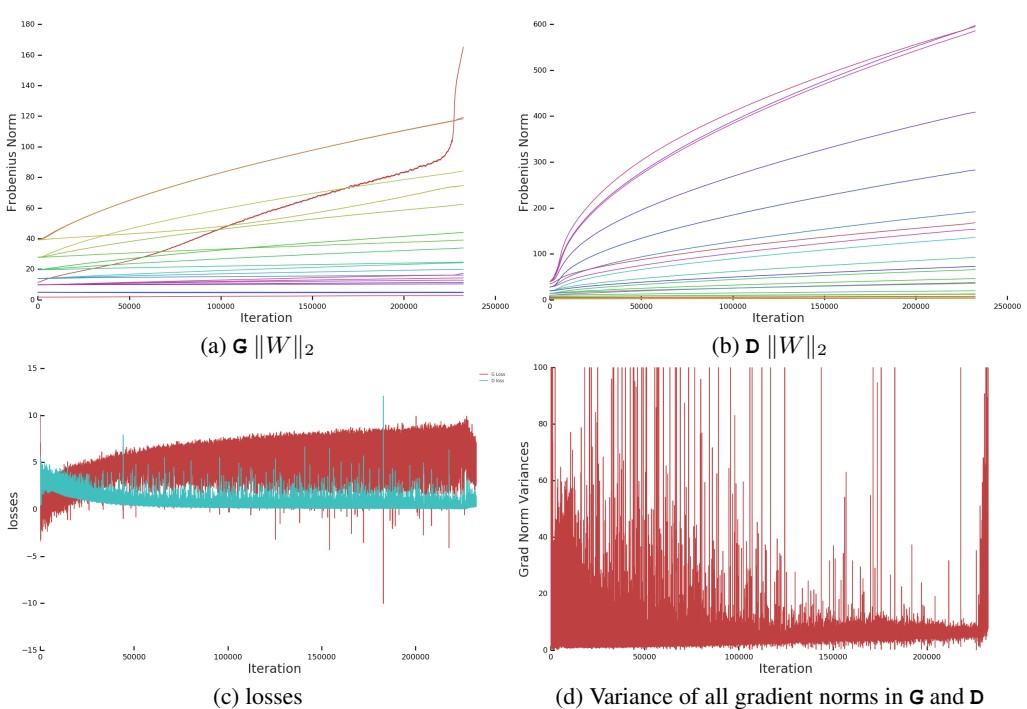

(a) **G** $\|W\|_2$      (b) **D** $\|W\|_2$

(c) losses      (d) Variance of all gradient norms in **G** and **D**

Figure 27: Additional training statistics for a typical model without special modifications. Collapse occurs after 200000 iterations.

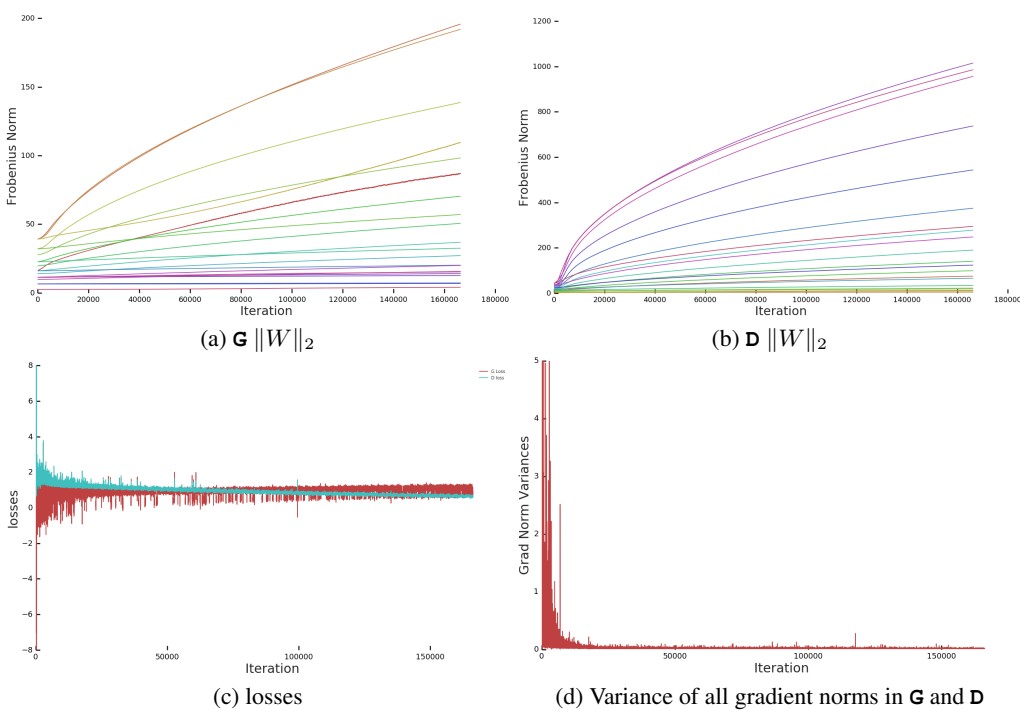

(a) **G** $\|W\|_2$      (b) **D** $\|W\|_2$

(c) losses      (d) Variance of all gradient norms in **G** and **D**

Figure 28: Additional training statistics with an R1 Gradient Penalty of strength 10 on **D**. This model does not collapse, but only reaches a maximum IS of 55.

## APPENDIX G  ADDITIONAL DISCUSSION: STABILITY AND COLLAPSE

In this section, we present and discuss additional investigations into the stability of our models, expanding upon the discussion in Section 4.

### G.1  INTERVENING BEFORE COLLAPSE

The symptoms of collapse are sharp and sudden, with sample quality dropping from its peak to its lowest value over the course of a few hundred iterations. We can detect this collapse when the singular values in **G** explode, but while the (unnormalized) singular values grow throughout training, there is no consistent threshold at which collapse occurs. This raises the question of whether it is possible to prevent or delay collapse by taking a model checkpoint several thousand iterations before collapse, and continuing training with some hyperparameters modified (e.g., the learning rate).

We conducted a range of intervention experiments wherein we took checkpoints of a collapsed model ten or twenty thousand iterations before collapse, changed some aspect of the training setup, then observed whether collapse occurred, when it occurred relative to the original collapse, and the final performance attained at collapse.

We found that increasing the learning rates (relative to their initial values) in either **G** or **D**, or both **G** and **D**, led to immediate collapse. This occurred even when doubling the learning rates from $2 \cdot 10^{-4}$ in **D** and $5 \cdot 10^{-5}$ in **G**, to $4 \cdot 10^{-4}$ in **D** and $1 \cdot 10^{-4}$ in **G**, a setting which is not normally unstable when used as the initial learning rates. We also tried changing the momentum terms (Adam's $\beta_1$ and $\beta_2$), or resetting the momentum vectors to zero, but this tended to either make no difference or, when increasing the momentum, cause immediate collapse.

We found that decreasing the learning rate in **G**, but keeping the learning rate in **D** unchanged could delay collapse (in some cases by over one hundred thousand iterations), but also crippled training— once the learning rate in **G** was decayed, performance either stayed constant or slowly decayed. Conversely, reducing the learning rate in **D** while keeping **G**'s learning rate led to immediate collapse. We hypothesize that this is because of the need for **D** to remain optimal throughout training—if its learning rate is reduced, it can no longer "keep up" with **G**, and training collapses. With this in mind, we also tried increasing the number of **D** steps per **G** step, but this either had no effect, or delayed collapse at the cost of crippling training (similar to decaying **G**'s learning rate).

To further illuminate these dynamics, we construct two additional intervention experiments, one where we freeze **G** before collapse (by ceasing all parameter updates) and observe whether **D** remains stable, and the reverse, where we freeze **D** before collapse and observe whether **G** remains stable. We find that when **G** is frozen, **D** remains stable, and slowly reduces both components of its loss towards zero. However, when **D** is frozen, **G** immediately and dramatically collapses, maxing out **D**'s loss to values upwards of 300, compared to the normal range of 0 to 3.

This leads to two conclusions: first, as has been noted in previous works (Miyato et al., 2018; Gulrajani et al., 2017; Zhang et al., 2018), **D** must remain optimal with respect to **G** both for stability and to provide useful gradient information. The consequence of **G** being allowed to win the game is a complete breakdown of the training process, regardless of **G**'s conditioning or optimization settings. Second, favoring **D** over **G** (either by training it with a larger learning rate, or for more steps) is insufficient to ensure stability even if **D** is well-conditioned. This suggests either that in practice, an optimal **D** is necessary but insufficient for training stability, or that some aspect of the system results in **D** not being trained towards optimality. With the latter possibility in mind, we take a closer look at the noise in **D**'s spectra in the following section.

## G.2 Spikes in the Discriminator's Spectra

(a) **D** $\sigma_0$          (b) **D** $\frac{\sigma_0}{\sigma_1}$

Figure 29: A closeup of **D**'s spectra at a noise spike.

If some element of **D**'s training process results in undesirable dynamics, it follows that the behavior of **D**'s spectra may hold clues as to what that element is. The top three singular values of **D** differ from **G**'s in that they have a large noise component, tend to grow throughout training but only show a small response to collapse, and the ratio of the first two singular values tends to be centered around one, suggesting that the spectra of **D** have a slow decay. When viewed up close (Figure 29), the noise spikes resemble an impulse response: at each spike, the spectra jump upwards, then slowly decrease, with some oscillation.

One possible explanation is that this behavior is a consequence of **D** memorizing the training data, as suggested by experiments in Section 4.2. As it approaches perfect memorization, it receives less and less signal from real data, as both the original GAN loss and the hinge loss provide zero gradients when **D** outputs a confident and correct prediction for a given example. If the gradient signal from real data attenuates to zero, this can result in **D** eventually becoming biased due to exclusively received gradients that encourage its outputs to be negative. If this bias passes a certain threshold, **D** will eventually misclassify a large number of real examples and receive a large gradient encouraging positive outputs, resulting in the observed impulse responses.

This argument suggests several fixes. First, one might consider an unbounded loss (such as the Wasserstein loss (Arjovsky et al., 2017)) which would not suffer this gradient attentuation. We found that even with gradient penalties and brief re-tuning of optimizer hyperparameters, our models did not stably train for more than a few thousand iterations with this loss. We instead explored changing the margin of the hinge loss as a partial compromise: for a given model and minibatch of data, increasing the margin will result in more examples falling within the margin, and thus contributing to the loss.[3]. Training with a smaller margin (by a factor of 2) measurably reduces performance, but training with a larger margin (by up to a factor of 3) does not prevent collapse or reduce the noise in **D**'s spectra. Increasing the margin beyond 3 results in unstable training similar to using the Wasserstein loss. Finally, the memorization argument might suggest that using a smaller **D** or using dropout in **D** would improve training by reducing its capacity to memorize, but in practice this degrades training.

---

[3]Unconstrained models could easily learn a different output scale to account for this margin, but the use of Spectral Normalization constrains our models and makes the specific selection of the margin meaningful.

## APPENDIX H   NEGATIVE RESULTS

We explored a range of novel and existing techniques which ended up degrading or otherwise not affecting performance in our setting. We report them here; our evaluations for this section are not as thorough as those for the main architectural choices.

Our intention in reporting these results is to save time for future work, and to give a more complete picture of our attempts to improve performance or stability. We note, however, that these results must be understood to be specific to the particular setup we used. A pitfall of reporting negative results is that one might report that a particular technique doesn't work, when the reality is that this technique did not have the desired effect when applied in a particular way to a particular problem. Drawing overly general conclusions might close off potentially fruitful avenues of research.

- We found that doubling the depth (by inserting an additional Residual block after every up- or down-sampling block) hampered performance.

- We experimented with sharing class embeddings between both G and D (as opposed to just within G). This is accomplished by replacing D's class embedding with a projection from G's embeddings, as is done in G's BatchNorm layers. In our initial experiments this seemed to help and accelerate training, but we found this trick scaled poorly and was sensitive to optimization hyperparameters, particularly the choice of number of D steps per G step.

- We tried replacing BatchNorm in G with WeightNorm (Salimans & Kingma, 2016), but this crippled training. We also tried removing BatchNorm and only having Spectral Normalization, but this also crippled training.

- We tried adding BatchNorm to D (both class-conditional and unconditional) in addition to Spectral Normalization, but this crippled training.

- We tried varying the choice of location of the attention block in G and D (and inserting multiple attention blocks at different resolutions) but found that at 128×128 there was no noticeable benefit to doing so, and compute and memory costs increased substantially. We found a benefit to moving the attention block up one stage when moving to 256×256, which is in line with our expectations given the increased resolution.

- We tried using filter sizes of 5 or 7 instead of 3 in either G or D or both. We found that having a filter size of 5 in G only provided a small improvement over the baseline but came at an unjustifiable compute cost. All other settings degraded performance.

- We tried varying the dilation for convolutional filters in both G and D at 128×128, but found that even a small amount of dilation in either network degraded performance.

- We tried bilinear upsampling in G in place of nearest-neighbors upsampling, but this degraded performance.

- In some of our models, we observed class-conditional mode collapse, where the model would only output one or two samples for a subset of classes but was still able to generate samples for all other classes. We noticed that the collapsed classes had embeddings which had become very large relative to the other embeddings, and attempted to ameliorate this issue by applying weight decay to the shared embedding only. We found that small amounts of weight decay ($10^{-6}$) instead degraded performance, and that only even smaller values ($10^{-8}$) did not degrade performance, but these values were also too small to prevent the class vectors from exploding. Higher-resolution models appear to be more resilient to this problem, and none of our final models appear to suffer from this type of collapse.

- We experimented with using MLPs instead of linear projections from G's class embeddings to its BatchNorm gains and biases, but did not find any benefit to doing so. We also experimented with Spectrally Normalizing these MLPs, and with providing these (and the linear projections) with a bias at their output, but did not notice any benefit.

- We tried gradient norm clipping (both the global variant typically used in recurrent networks, and a local version where the clipping value is determined on a per-parameter basis) but found this did not alleviate instability.

## APPENDIX I  HYPERPARAMETERS

We performed various hyperparameter sweeps in this work:

- We swept the Cartesian product of the learning rates for each network through $[10^{-5}, 5 \cdot 10^{-5}, 10^{-4}, 2 \cdot 10^{-4}, 4 \cdot 10^{-4}, 8 \cdot 10^{-4}, 10^{-3}]$, and initially found that the SA-GAN settings (`G`'s learning rate $10^{-4}$, `D`'s learning rate $4 \cdot 10^{-4}$) were optimal at lower batch sizes; we did not repeat this sweep at higher batch sizes but did try halving and doubling the learning rate, arriving at the halved settings used for our experiments.

- We swept the R1 gradient penalty strength through $[10^{-3}, 10^{-2}, 10^{-1}, 0.5, 1, 2, 3, 5, 10]$. We find that the strength of the penalty correlates negatively with performance, but that settings above $0.5$ impart training stability.

- We swept the keep probabilities for DropOut in the final layer of `D` through $[0.5, 0.6, 0.7, 0.8, 0.9, 0.95]$. We find that DropOut has a similar stabilizing effect to R1 but also degrades performance.

- We swept `D`'s Adam $\beta_1$ parameter through $[0.1, 0.2, 0.3, 0.4, 0.5]$ and found it to have a light regularization effect similar to DropOut, but not to significantly improve results. Higher $\beta_1$ terms in either network crippled training.

- We swept the strength of the modified Orthogonal Regularization penalty in `G` through $[10^{-5}, 5 \cdot 10^{-5}, 10^{-4}, 5 \cdot 10^{-4}, 10^{-3}, 10^{-2}]$, and selected $10^{-4}$.

