# OpenReview forum: "Large Scale GAN Training for High Fidelity Natural Image Synthesis"
_ICLR.cc/2019/Conference_

### Official Review · AnonReviewer3 · 2018-10-26
**Great progress achievement in the field of image generation**

**Rating:** 9
**Confidence:** 4

**Review:**

This paper present extensions of the Self-Attention Generative Adversarial Network approach SAGAN, leading to impressive images generations conditioned on imagenet classes.
The key components of the approach are :
- increasing the batch size by a factor 8
- augmenting the width of the networks by 50%
These first two elements result in an Inception score (IS) boost from 52 to 93.
- the use of shared embeddings for the class conditioned batch norm layers, orthonormal regularization and hierarchical latent space bring an additional boost of IS 99.
The core novel element of the paper is the truncation trick: At train time, the input z is sampled from a normal distribution but at test time, a truncated normal distribution is used: when the magnitude of elements of z are above a certain threshold, they are re-sampled.
Variations of this threshold lead to variations in FD and IS, as shown in insightful experiments. The comments that more data helps (internal dataset experiments) is also informative.
Very nice to have included negative results and detailed parameter sweeps.

This is a very nice work with impressive results, a great progress achievement in the field of image generation.
Very well written.

Suggestions/questions:
- it would be nice to also propose unconditioned experiments.
It would be good to give an idea in the text of TPU-GPU equivalence in terms of feasibility of a standard GPU implementation - computation time it would involve.
- I understand that no data augmentation was used during training?
- clarification of the truncation trick: if the elements of z are re-sampled and are still above the threshold, are they re-sampled again and again until they are all below the given threshold?
- A sentence could be added to explain the truncation trick in the abstract directly since it is simple to understand and is key to the quality of the results.
- A reference to Appendix C could be given at the beginning of the Experiments section to help the reader find these details more easily.
- It would be nice to display more Nearest neighbors for the dog image.
- It would be nice to add a figure of random generations.
- make the bib uniform: remove unnecessary doi - url - cvpr page numbers

---

> ### Author Response · Authors · 2018-11-21
> **Response to Reviewer 3**
>
> We would like to thank Reviewer 3 for the review and constructive suggestions. Our responses inline:
>
> >it would be nice to also propose unconditioned experiments.
> -We agree; this was simply not within the scope of the work we conducted.
>
> >I understand that no data augmentation was used during training?
> -This is correct, and consistent with previous works (Spectral Normalization and WGAN-GP). We briefly experimented with data augmentation (random crops and horizontal flips) but did not notice any measurable performance difference.
>
> >clarification of the truncation trick: if the elements of z are re-sampled and are still above the threshold, are they re-sampled again and again until they are all below the given threshold?
> -Yes, this can effectively be seen as modifying the PDF of z to have no mass outside of the truncation threshold. TensorFlow offers a built-in implementation with tf.random.truncated_normal.
>
> >A sentence could be added to explain the truncation trick in the abstract directly since it is simple to understand and is key to the quality of the results.
> -We have revised the abstract to explain the truncation trick as controlling the tradeoff between fidelity and diversity by reducing the variance of the Generator’s input.
>
> >A reference to Appendix C could be given at the beginning of the Experiments section to help the reader find these details more easily.
> -Thanks for the pointer! We have added this reference.
>
> >It would be nice to add a figure of random generations.
> -In the caption of Figure 5, we include a link to an anonymous drive folder with sample sheets at different resolutions and truncation values, with 12 random images per class.
>
> >make the bib uniform: remove unnecessary doi - url - cvpr page numbers
> -Thanks, we have fixed this.

---

### Official Review · AnonReviewer2 · 2018-10-31
**Good investigation, great results, could be improved.**

**Rating:** 7
**Confidence:** 3

**Review:**

Summary:
The authors present a empirical investigation of methods for scaling GANs to complex datasets, such as ImageNet, for class-conditioned image generation. They first build and describe a strong baseline based on recently proposed techniques for GANs and push the performance on large datasets with several modifications presented sequentially, to obtain strong state-of-the-art IS/FID scores, as well as impressive visual results. The authors propose a simple truncation trick to control the fidelity/variance which is interesting on its own but cannot always scale with the architecture. The authors further propose a orthogonalization-based regularization to mitigate this problem. An investigation of training collapse at large scale is also performed; the authors investigate some regularization schemes based on gathered empirical evidence. As a result, they explore and discard Spectral Normalization of the generator as a way to prevent collapse and show that a severe tradeoff between stability and quality can be controlled when using zero-centered gradient penalties in the Discriminator. In the end, no solution that can ensure quality and stability is found, except having prohibitively large amounts of data (~300M images). Models are evaluated on the ImageNet and on this internal, bigger dataset.

Pros:
- This investigation gives a significant amount of insights on GAN stability and performance at large scales, which should be useful for anyone working with GANs on complex datasets (and that have access to great computational resources).

- Even though commonly used evaluations metrics for GANs are still not fully adequate, the authors obtain quantitative performance significantly beyond previous work, which seems indeed correlated with remarkable visual results.

- The baseline and added modifications are well presented and clearly explained. The Appendices also have great value in that regard.


Cons:
- Discussions sometimes lack depth or are absent.
For example, it is unclear to me why some larger models are not amenable to truncation. Besides visible artifacts, what does it mean? Why does a smoother G reduces those artifacts? Were samples from those networks better without using truncation? Why would this be?

Authors report how wider networks perform best, and how deeper networks degrade performance. Again, discussions are lacking, and it doesn’t seem the authors tried to understand why such behaviors were shown.

Even though this is mostly an empirical investigation, I think some more efforts should be put in understanding and explaining why some of those behaviors are shown, as I think it can bootstrap future work more easily.

- In Section 3.1 : “Across runs in Table 1, we observe that without Orthogonal Regularization, only 16% of models are amenable to truncation compared to 60% with Orthogonal Regularization.” For me, this is not particularly clear. Is this something the reader should understand from Table 1?

- I question the choice of sections chosen to be in the main paper/appendices. I greatly appreciated the negative results reported in the main text as well as in the appendices and this has significant value. However, as this is to me mostly a detailed empirical investigation and presentation of high-performance GANs on large scales, I would be likely to share this with colleagues who want to tackle similar problems. In this case, if future readers limit themselves to the main text, I think it can have more value to present some content form Appendix B and C than to have more than a full page on stability investigations and attempted tricks that turned out not to be used to reach maximal performance. However I do not want to discourage publishing of negative results, and I definitely wish to see this investigation in the paper, but I merely question the positioning of such information. With regard to my first negative point above about the lack of discussions, it seems the analysis of Section 4 is disproportionate compared to other places.


Suggestions/Comments:

- Regarding the diversity/fidelity tradeoff using different truncation thresholds, I think constraining the norm of the sampled noise vectors to the exact threshold value (by projecting the samples on the 0-centered hyper-sphere of radius = threshold) could yield even more interesting or more informative Figures, as obtained scores or samples on the edge of that hyper-sphere might provide information on the ‘guaranteed’ (not proven) quality/fidelity of samples mapped from inside that hyper-sphere.

- In Appendix D, the Figures could be slightly clarified by using a colored heatmap to color the curve, with colors corresponding to the threshold values. Similar curves could also be produced with the hyper-sphere projection proposed above to have a slightly clearer idea of the behavior on the limit of that hyper-sphere.

- In Section 4.2, in the second paragraph, you refer to Appendix F and describe “sharp upward jump at collapse” in D’s loss. However, it seems the only Figure showing D’s loss when unconstrained is Figure 26, in which it is hard to notice any significant jump in the loss.

- In Appendix F, Figure 20 d), the title seems wrong. It seems to report sigma^2 values, but the title says “losses”.


This investigation of GAN scalability is successful results-wise even though the inability to stabilize training without sacrificing great performance on ImageNet is disappointing. The improvement over previous SOTA is definitely significant. This work thus shows a modern GAN architecture for complex datasets that could be a strong basis for future work. However, I think the paper could and should be improved with some more detailed analysis and discussions of exhibited behaviors in order to further guide and encourage future work. It could also be clarified on some aspects, and potentially re-structured a bit to be better align with its probable impact directions.  I would also be curious to see the proposed techniques applied on simpler datasets. Can this be useful for someone having less compute power and working on something similar to CelebA?

---

> ### Author Response · Authors · 2018-11-21
> **Response to Reviewer 2 (Part 1/2)**
>
> We would like to thank Reviewer 2 for their review and constructive suggestions. Our responses inline:
>
> >Discussions sometimes lack depth or are absent.
> -We have added an additional section (Appendix G) expanding on our discussion and providing additional insight into the observed instabilities.
>
> >For example, it is unclear to me why some larger models are not amenable to truncation. Besides visible artifacts, what does it mean? Why does a smoother G reduces those artifacts?
> -Truncation introduces a train-test disparity in G’s inputs--at sampling time, G is given a distribution it has effectively never seen in training. The observation that imposing orthogonality constraints improves amenability to truncation is empirical. Our suspicion is that if G is not encouraged to be “smooth” in some sense, then it is likely that G will only properly generate images given points from the untruncated distribution. We hypothesize that models which are not amenable end up learning mappings which, when given truncated noise, either attenuate or amplify certain activation pathways, leading to extreme output values (hence the observed saturation artifacts). We speculate that encouraging G’s filters to have minimum pairwise cosine similarity means that, when exposed to distribution shift, the network’s features are less correlated and less likely to align and amplify an activation path it would otherwise have learned to scale properly.
>
>
> >Were samples from those networks better without using truncation? Why would this be?
> -Samples from those networks without truncation do not have measurably different quality, and their training metrics (losses, singular values) show no differences. Aside from empirically testing each network individually for amenability to truncation, we found no other way to check for that amenability.
>
> > Authors report how wider networks perform best, and how deeper networks degrade performance. Again, discussions are lacking, and it doesn’t seem the authors tried to understand why such behaviors were shown. Even though this is mostly an empirical investigation, I think some more efforts should be put in understanding and explaining why some of those behaviors are shown, as I think it can bootstrap future work more easily.
> -We are wary of explanations for which we do not have evidence. For each of the modifications introduced in Section 3, we offer a succinct conjecture as to why that change improves performance, but we are not aware of any existing reliable, informative metric which we could employ to understand or trace the source of each observed behavior, particularly with respect to GAN stability or performance.
> Regarding depth vs width: This paper is empirical, and we only briefly experimented with increasing depth analogously to increasing width. While increasing width provided an immediate measurable benefit, increasing depth did not. We felt that it was better to report the results of this brief investigation than to omit it for a lack of investigatory depth.

---

> > ### Author Response · Authors · 2018-11-21
> > **Response to Reviewer 2 (Part 2/2)**
> >
> > > In Section 3.1 : “Across runs in Table 1, we observe that without Orthogonal Regularization, only 16% of models are amenable to truncation compared to 60% with Orthogonal Regularization.” For me, this is not particularly clear. Is this something the reader should understand from Table 1?
> > -This means that of all the models we trained for the study presented in Table 1 which did not use Orthogonal Regularization, only 16% were amenable to truncation. Of all the models which we trained for the study presented in Table 2 which did use Orthogonal Regularization, 60% were amenable to truncation. This is not reflected in Table 1, which is merely a presentation of how the introduced modifications impact performance.
> >
> > >I question the choice of sections chosen to be in the main paper/appendices. I greatly appreciated the negative results reported in the main text as well as in the appendices and this has significant value. However, as this is to me mostly a detailed empirical investigation and presentation of high-performance GANs on large scales, I would be likely to share this with colleagues who want to tackle similar problems. In this case, if future readers limit themselves to the main text, I think it can have more value to present some content form Appendix B and C than to have more than a full page on stability investigations and attempted tricks that turned out not to be used to reach maximal performance. However I do not want to discourage publishing of negative results, and I definitely wish to see this investigation in the paper, but I merely question the positioning of such information. With regard to my first negative point above about the lack of discussions, it seems the analysis of Section 4 is disproportionate compared to other places.
> > -We appreciate this suggestion. While we recognize that this paper generally has a strong focus on implementation details, we felt that this instability was one of the most salient behaviors we observed, and that future work would be best served by presenting our investigations and attempts to understand its source, even if these methods did not improve performance.  The information in Appendix B and C is intended to be of interest to those who want to reproduce our experiments, so it largely comprises hyperparameters and architectural details that we felt were not necessary to understand the main results of the paper.
> >
> > >In Appendix F, Figure 20 d), the title seems wrong. It seems to report sigma^2 values, but the title says “losses”.
> > -Thanks! This was indeed an error, which we’ve corrected in the updated draft.
> >
> > >I would also be curious to see the proposed techniques applied on simpler datasets. Can this be useful for someone having less compute power and working on something similar to CelebA?
> > -The goal of this work is to explore GANs at large scale; the exploration of small or medium scale models would indeed be interesting for another study. Having said that, we do evaluate BigGAN on conditional CIFAR-10 (mentioned briefly in Appendix C.2) and obtain an IS of 9.22 and an FID of 14.73 without truncation, which to our knowledge are better than any published results.

---

### Official Review · AnonReviewer1 · 2018-11-02
**Good paper**

**Rating:** 8
**Confidence:** 4

**Review:**

Summary:
This paper proposes a suite of tricks for training large-scale GANs, and obtaining state-of-the-art results for high-resolution images. The paper starts from a self-attention GAN baseline (Zhang 2018), and proposes:
-	Increasing batch size (8x) and model size (2x)
-	Splitting noise z in multiple chunks, and injecting it in multiple layers of the generator
-	Sampling from truncated normal distribution, where samples with norms that exceed a specific threshold are resampled. This seems to be used only at test-time and is used to control variety-fidelity tradeoff. The generator is encouraged to be smooth using an orthogonal regularization term.
In addition, the paper proposes practical recipes for characterizing collapse in GANs. In the generator, the exploding of the top 3 singular values of each weight matrix seem to indicate collapse. In the discriminator, the sudden increase of the ratio of first/second singular value of weight matrices indicate collapse in GANs. Interestingly, the paper suggests that various regularization methods which can improve stability in GAN training, do not necessarily correspond to improvement in performance.

Strengths:
-	Proposed techniques are intuitive and very well motivated
-	One of the big pluses of this work is that authors try to "quantify" each proposed technique with training speed and/or performance improvement. This is really a good practice.
-	Detailed analysis for detecting collapse and improving stability in large-scale GAN
-	Probably no need to mention that, but results are quite impressive

Weaknesses:
-	Computational budget required is massive. The paper mentions model use from 128-256 TPUs, which severely limits reproducibility of results.

Comments/Questions:
-	Can you elaborate more on why BatchNorm statistics are computed across all devices as opposed to per-device? Was this crucial for best performance?
-	It is not clear if provided analysis for large-scale GANs apply for small-medium sized GANs. Providing such analysis would be also helpful for the community.
-	How do you see the impact of the suggested techniques on tackling harder data-modalities for GANs, e.g. text or sequential data in general?

Overall recommendation:
The paper is well written, ideas are well motivated/justified and results are very compelling. This is a good paper and I higly recommend acceptance.

---

> ### Author Response · Authors · 2018-11-21
> **Response to Reviewer 1**
>
> We would like to thank Reviewer 1 for their review and constructive suggestions. Our responses inline:
>
> >Can you elaborate more on why BatchNorm statistics are computed across all devices as opposed to per-device? Was this crucial for best performance?
> -The primary reason is to ensure that training is invariant to the per-device batch size. When scaling from resolution 128x128 to 256x256, we increase the number of devices but maintain the same overall batch size, reducing the per-device batch size. Cross-replica BatchNorm ensures that the smaller per-device batch size does not affect training. Switching to per-device BatchNorm at 128x128 results in a performance drop, albeit not a crippling one: for a model which would otherwise get an IS of 92 and an FID of 9.5, switching to per-device BatchNorm results in an IS of 78 and FID of 13.
>
> >It is not clear if provided analysis for large-scale GANs apply for small-medium sized GANs. Providing such analysis would be also helpful for the community.
> -The goal of this work is to explore GANs at large scale; the exploration of small or medium scale models would indeed be interesting for another study. Having said that, we do evaluate BigGAN on conditional CIFAR-10 (mentioned briefly in Appendix C.2) and obtain an IS of 9.22 and an FID of 14.73 without truncation, which to our knowledge are better than any published results.
>
> >How do you see the impact of the suggested techniques on tackling harder data-modalities for GANs, e.g. text or sequential data in general?
> -Any of the proposed techniques could be applied to standard GANs for text or other sequential data in principle, but we have not experimented with these applications ourselves.

---

### Public Comment · ~Mert_Bülent_Sarıyıldız1 · 2018-09-30
**Regarding Conditional Batchnorms**

Thank you for all your efforts towards understanding training dynamics in large-scale GANs. I have a question about conditional batch-norms.  You mention in Section 3 these
* " Instead of having a separate layer for each embedding (Miyato et al., 2018; Zhang et al., 2018), we opt to use a shared embedding, which is linearly projected to each layer’s gains and biases (Perez et al., 2018)."
* "For our architecture, this is easily accomplished by splitting z into one chunk per resolution, and concatenating each chunk to the conditional vector c which gets projected to the BatchNorm gains and biases. ".

I believe that these statements need more clarification. i) how do you define a chunk?, ii) How z is split into chunks? iii) How do you compute shared embedding? iv) how parameters of an affine transformation for each layer is constructed from the shared embedding?

Regards.

---

> ### Author Response · Authors · 2018-09-30
> **response to Mert**
>
> Hi Mert,
>
> i/ii):
> Please see our appendix for further details. A chunk refers to a subset of the dimensions of z in the channel dimension; if z is a 100 x 128-dimensional tensor (batch size x channels) sampled from N(0,1), then splitting it into 8 chunks would result in 8 tensors (z_i for i=1 to 8 )  each of dimension 100 x 16.  E.g.
> z = tf.random_normal((100,128))
> z_chunks = tf.split(z, 8, axis=1)
>
> iii / iv):
> In previous works on conditional GANs, the conditional batchnorm gains and biases are implemented as embeddings, similar to word embeddings in language models, with one embedding per layer.  We replace this with a single embedding which we pass through a single linear transform to get the batchnorm parameters. We describe this in the appendix, but here's some pseudocode:
> embedding_weights  = matrix in (num_classes, embedding_dimension)
> bias_projection = matrices in (embedding_dimension, batchnorm_channels_dimension)
> gain_projection = matrices in (embedding_dimension, batchnorm_channels_dimension)
>
> shared_embedding = embedding_weights * one_hot(class index)
> bias_i = bias_projection_i * shared_embedding
> gain_i = 1 + gain_projection_i * shared_embedding
>
> If you're using hierarchical latents, use this instead:
> bias_i = bias_projection_i * concatenate(shared_embedding,  z_chunks_i)
> gain_i = 1 + gain_projection_i *concatenate(shared_embedding,  z_chunks_i)
>
> Hope that helps!

---

> > ### Public Comment · (anonymous) · 2019-01-31
> > **embedding_dimension**
> >
> > Hi, I have a follow-up question regarding condition Batchnorm. What are the values of the embedding_dimension used for G with different resolutions? I could not find this information in the paper. Thanks.

---

### Public Comment · ~Sheng_Hu1 · 2018-10-01
**Question about Inception Score of ImageNet validation set**

I have a question about Inception Score. You mention in APPENDIX C "We compute the IS for both the training and validation sets. At 128×128 the training data has an IS of 233, and the validation data has an IS of 166..."
However, in Table 1 of "A Note on the Inception Score", which is referenced by your paper, the Inception Score of ImageNet validation set is around 63.
I wonder what is the cause of the gap between these two scores.

---

> ### Author Response · Authors · 2018-10-01
> **Response to Sheng**
>
> Hi Sheng,
>
> The score reported in "A note on the Inception Score"  is for ImageNet at 64x64 resolution. We get approximately the same number using our code.
>
> Thanks.

---

### Public Comment · ~Jaonary_Rabarisoa1 · 2018-10-03
**Question about the architecture and the scalability**

One most striking results of your paper is the effect of the batch size. In your experiment you use some TPU cores so I guess that you have enough memory to store all of your batch. Do you think that it is possible to get the same result if you use multiple GPUs instead with reduced batch size and algorithm such as all reduce to aggregate the gradients ?
One more thing, it's not really clear what is the difference of your architecture and the one used by Miyato 2018 ? You said in the appendix B that the number of filters of the first conv layer of each block is equal to the number of the output filters but not the number of the input filters. Can you explain better what does it mean ?

---

> ### Author Response · Authors · 2018-10-03
> **Response to Jaonary**
>
> Hi Jaonary,
>
> It may be possible to get similar results using gradient aggregation, but it's tough to say--we use cross-replica BatchNorm in the Generator, so aggregating gradients with a smaller batch size will not be exactly equivalent. In our ablations using per-device BatchNorm reduced performance but still trained, so perhaps aggregating gradients with cross-replica BatchNorm and multiple GPUs will work (albeit it will be quite slow and not exactly equivalent to what we've done).
>
> The architectural difference is in the channel pattern of the Discriminator, where each residual block takes in a tensor with num_in channels and outputs a tensor with num_out channels. In (Miyato, 2018) the first convolution in the residual block has num_in outputs, and the second convolution has num_out outputs. In (Zhang, 2018), however, the first convolution in the residual block has num_out outputs instead of num_in inputs, which results in the Discriminator having more parameters and more capacity. We use the channel pattern from (Zhang, 2018).
>
> Thanks.

---

### Public Comment · ~kohei_nishimura1 · 2018-10-04
**Question about saturation artifacts**

I have a question about saturation artifacts you mention in section 3.

>> The distribution shift caused by sampling with different latents than those seen in training is problematic for many models.
>> Some of our larger models are not amenable to truncation, producing saturation artifacts (Figure 2(b)) when fed truncated noise.

I wonder why larger models produce saturation artifacts when fed truncated noise.
the noise outside the range can be sampled from N(0, 1). so, I believe modles produce saturation artifacts without truncated trick.
I believe that the reason why truncate trick produce saturation artifacts need more clarification.

Regards.

---

### Public Comment · (anonymous) · 2018-10-18
**Question about singular values**

Thank you for great insights into instabilities of the generator and discriminator! I'm a little bit confused though. Do you employ Spectral Normalization in the generator and discriminator? Spectral normalization should make the largest singular value of the weight matrix around 1 but Figure 3 shows very large eigenvalues. Am I missing something?

---

> ### Author Response · Authors · 2018-10-18
> **Response to Anonymous**
>
> Hi,
>
> As mentioned in the first paragraph of Section 3, we use Spectral norm in both G and D. As mentioned in the caption of Figure 3, the spectra we plot are before spectral normalization, so the actual values will be normalized by the first singular value. We plot the unnormalized values to show how the spectra of the underlying weights change over time.
>
> Thanks.

---

> > ### Public Comment · (anonymous) · 2018-10-18
> > **Follow-up**
> >
> > Thank you for the clarification! It's very interesting that pre-SN singular values of some layers' weight keeps growing. That seems to suggest that the outputs of the layer lie in a very low-dimensional subspace.

---

### Author Response · Authors · 2018-10-30
**Relevant Prior Work**

We've recently been made aware of two prior works that observe a correlation between the variance of the latent noise and the variety/quality of the Generator outputs. We will be adding references accordingly.

[1] Marco Marchesi. Megapixel Size Image Creation using Generative Adversarial Networks. arXiv preprint arXiv:1706.00082.
[2] Mathijs Pieters and Marco Wiering. Comparing Generative Adversarial Network Techniques for Image Creation and Modification. arXiv preprint arXiv:1803.09093.

---

### Public Comment · (anonymous) · 2018-11-05
**Question about the architecture**

Hi, I would like to ask some more details about your architecture.

1. Do you apply spectral normalization in the attention layer (non-local block)?

2. Do you apply batch normalization in the discriminator?

3. Do you perform batch normalization or nonlinear (relu) to the input of the attention layer (non-local block) before transforming the input in to the feature spaces f, g?

4. In the non-local block, the weight gamma is initialized as 0. Did you observe that gamma becomes negative during training? Or did you force gamma to be non-negative?

Thanks!

---

> ### Author Response · Authors · 2018-11-05
> **Response to Question about Architecture**
>
> 1. Yes, the non-local blocks have spectral normalization applied to the convolutional weights, as in SA-GAN.
>
> 2. No, following SN-GAN there is no BatchNorm in D.
>
> 3. We do not apply BatchNorm or ReLU before the non-local block--it takes in the output of the previous residual block. Please see https://github.com/brain-research/self-attention-gan for a reference implementation of non-local blocks.
>
> 4. The sign of gamma is arbitrary (the output of the block before being multiplied by gamma can take on either sign), and we observe both positive and negative gammas in our models. Gamma is a freely learned scalar parameter.
>
> Thanks.

---

> > ### Public Comment · (anonymous) · 2018-11-05
> > **Thanks**
> >
> > Thanks for sharing the details!

---

### Public Comment · (anonymous) · 2018-11-05
**Question about unsupervised training**

The samples look extremely good. Have you tried to calculate intra-class FID like the cGANs with Projection Discriminator did? Also, have you tried training your model on any unlabelled data set?

---

### Public Comment · (anonymous) · 2018-11-06
**Question about calculating inception scores**

Hi, I would like to ask some details about calculating inception scores.

How did you calculate the inception score for images of 128x128 and 512x512 resolutions?
Did you just resize the images to 229x229 and feed them into the inception-v3 model, which was pretrained on 229x229 imagenet dataset?

For calculating IS, did you use the code provided by openai?
https://github.com/openai/improved-gan

Thanks in advance.

---

### Author Response · Authors · 2018-11-21
**Draft Update**

We would like to thank all reviewers for their reviews. We have uploaded a revised draft incorporating this feedback. Specifically:
-We have added reference to the two papers mentioned in an earlier comment, as well as “The Unusual Effectiveness of Averaging in GAN Training,” Yazici et al., arXiv:1806.04498.
-Added Appendix G expanding on our discussion, and referenced this appendix at the end of section 4.2
-Fixed typos in captions
-Added a brief section on pitfalls of negative results in negative results appendix

---

### Meta-Review · Area_Chair1 · 2018-12-08
**New SOTA for image sampling**

**Confidence:** 4
**Recommendation:** Accept (Oral)

**Metareview:**

The paper proposes a set of tricks leading to a new SOTA for sampling high resolution images. It is clearly written and the presented contribution will be of high interest for practitioners.